



# Using [226]Ra and [228]Ra isotopes to distinguish water mass distribution in the Canadian Arctic Archipelago

Chantal Mears[1], Helmuth Thomas[1,2*], Paul B. Henderson[3], Matthew A. Charette[3], Hugh MacIntyre[1], Frank Dehairs[4], Christophe Monnin[5], and Alfonso Mucci[6]

[1]: Dalhousie University, Department of Oceanography, Halifax, NS, Canada

[2]: Helmholtz-Centre-Geesthacht, Institute for Coastal Research, Geesthacht, Germany.

[3]: Woods Hole Oceanographic Institution, Department of Marine Chemistry and Geochemistry, Woods Hole, MA, USA.

[4]: Earth System Sciences and Analytical and Environmental Chemistry, Vrije Universiteit Brussel, Brussels, Belgium.

[5]: CNRS -Université Paul Sabatier-IRD-OMP, Geosciences Environnement Toulouse (GET), 14 Avenue Edouard Belin, 31400 Toulouse FRANCE

[6]: GEOTOP and Department of Earth and Planetary Sciences, McGill University, Montréal, QC, Canada

[*]: corresponding author, email: helmuth.thomas@hzg.de





**Abstract:**

As a shelf dominated basin, the Arctic Ocean and its biogeochemistry are heavily influenced by continental and riverine sources. Radium isotopes ($^{226}$Ra, $^{228}$Ra, $^{224}$Ra, $^{223}$Ra), are transferred from the

sediments to seawater, making them ideal tracers of sediment-water exchange processes and ocean mixing. $^{226}$Ra and $^{228}$Ra are the two longer-lived isotopes of the Radium Quartet ($^{226}$Ra, $t_{1/2}$=1600y and $^{228}$Ra, $t_{1/2}$=5.8y). Because of their long half-lives they can provide insight into the water mass compositions, distribution patterns, as well as mixing processes and the associated timescales throughout the Canadian Arctic Archipelago (CAA). The wide range of $^{226}$Ra, $^{228}$Ra, and of the

$^{228}$Ra/$^{226}$Ra ratio, measured in water samples collected during the 2015 GEOTRACES cruise, complemented by additional chemical tracers (dissolved inorganic carbon (DIC), total alkalinity (AT), barium (Ba), and the stable oxygen isotope composition of water ($\delta^{18}$O)) highlight the dominant biogeochemical, hydrographic and bathymetric features of the CAA. Bathymetric features, such as the continental shelf and shallow coastal sills, are critical in modulating circulation patterns within the

CAA, including the bulk flow of Pacific waters and the inhibited eastward flow of denser Atlantic waters through the CAA. Using a Principal Component Analysis, we unravel the dominant mechanisms and the apparent water mass end-members that shape the tracer distributions. We identify two distinct water masses located above and below the upper halocline layer throughout the CAA, as well as distinctly differentiate surface waters in the eastern and western CAA. Furthermore, we identify water

exchange across 80°W, inferring a draw of Atlantic water, originating from Baffin Bay, into the CAA. In other words, this implies the presence of an Atlantic water U-turn located at Barrow Strait, where the same water mass is seen along the northernmost edge at 80°W as well as along south-easternmost confines of Lancaster Sound. Overall, this study provides a stepping stone for future research initiatives within the Canadian Arctic Archipelago, revealing how quantifying disparities in radioactive isotopes





can provide valuable information on the potential effects of climate change within vulnerable areas such as the CAA.

## I: Introduction

### I.I. General Background

Over the past 30 years, major research initiatives have been undertaken within the Arctic, highlighting this region's global importance and vulnerability to climate change (Prinsenberg and Bennett, 1987; Shadwick et al., 2013). One of the primary causes of this vulnerability is a modification of the regional hydrographic regime, characterized by cool, $CO_2$-charged (less alkaline) Pacific waters, that enter the Arctic Ocean via the Bering Strait, flowing along the southern parts of the Canadian Arctic Archipelago (CAA) and being dispersed into Baffin Bay. Previous studies have shown that these eastward flowing waters contribute significantly to carbon sequestration as well as instigate deep-water formation in the North Atlantic (e.g., Aagaard and Carmack, 1989; Burt et al., 2016a; Curry et al., 2011; Hamilton and Wu, 2013; Holland et al., 2001; Ingram and Prinsenberg, 1998; Rahmstorf, 2002; Shadwick et al., 2011a).

Although the various water masses delivered to Baffin Bay play a role in establishing and maintaining the global thermohaline circulation, little is known about the distribution, composition, and modes of delivery of water through the Canadian Archipelago. This study contributes to the knowledge base of circulation patterns in the CAA by using the radioactive radium isotopes $^{228}$Ra and $^{226}$Ra as well as dissolved inorganic carbon (DIC), total alkalinity (AT), barium (Ba), and the stable oxygen isotope composition of water ($\delta^{18}$O) as tracers of water mass distribution, mixing, and composition throughout the region. Moreover, we hope that this study will provide a platform for further investigations of how changes in environmental conditions within this vulnerable area will affect the distribution of these tracers, as well as biogeochemical cycles and circulation in the CAA.






### I.II. Oceanographic Setting

Approximately 30-50% of the Arctic Ocean surface area (totaling to $9.5 \times 10^6 km^2$) is dominated by polar continental shelves (Coachman and Aagaard, 1974; Jakobsson, 2002; Rutgers van der Loeff et al., 1995; Shadwick et al., 2011b; Walsh, 1991; Xing et al., 2003). The CAA, a region of branching

channels and straits that extends from approximately 120°W to 80°W is located in this shelf-dominated region (Fig. 1). Spanning only 65km across at its widest, this narrow, polar network provides a critical connection between the Pacific and Atlantic Oceans and facilitates the export of approximately one third of the Arctic Ocean's outflowing water (Coachman and Aagaard, 1974; Hamilton et al., 2013; Hamilton and Wu, 2013).

Previous research has partitioned the water column in the CAA into three salinity-defined water masses, the deepest and most saline being the Atlantic Layer (ATL, $S_p > 33.1$), followed by the Pacific Upper Halocline Layer (UHL; $31 < S_p < 33.1$), and finally the least saline and uppermost being the Polar Mixed Layer (PML; $S_p < 31$) (e.g., Aagaard et al., 1981; Aagaard and Carmack, 1994; Bauch et al., 1995; Mathis et al., 2005; Shadwick et al., 2011a). All three water masses have been identified at

both the eastern and western boundaries of the CAA whereas only the upper two layers (PML and UHL) are found throughout. The presence of a 200 m shoal at Barrow Strait (Fig. 1), that bridges the western and eastern regions, prevents the Deep ATL water mass from flowing eastward through the CAA (Jones, 2003; Macdonald et al., 1987; Newton and Coachman, 1974). As a result, the bulk eastward transport is composed of the cool, $CO_2$-charged (less alkaline) Pacific and fresh surface

waters that flow from the Canada Basin to Baffin Bay through the CAA (Hamilton and Wu, 2013; Prinsenberg et al., 2009; Wang et al., 2012; Xing et al., 2003).

In addition to the bulk eastward transport through the CAA, the northern regions of the CAA host an occasional westward flowing current during late summer (Peterson et al., 2012; Prinsenberg & Bennett,





1987; Prinsenberg et al., 2009; Rudels, 1985). This suggests that there may be the intrusion of Atlantic

waters originating from Baffin Bay, moving into the CAA from the east, and possibly creating a "U-turn", rerouting westward current back into Baffin Bay. The importance of this "U-turn" will be discussed later in the results section.

    I.III. Some considerations about the two long-lived Radium Isotopes

$^{226}$Ra and $^{228}$Ra are the two longer-lived isotopes of the Radium Quartet ($^{226}$Ra, $t_{1/2}$=1600 y and $^{228}$Ra, $t_{1/2}$=5.8 y). They are found at readily detectable activities that are largely unperturbed by biological activity in seawater, allowing them to be used as nearly conservative radioactive tracers (Charette et al., 2015, 2016; IAEA, 2011; Moore et al., 1980). Additionally, both long-lived Ra isotopes are formed from the decay of different Thorium (Th) isotopes in sediments and are distributed to the ocean through

porewater advection and diffusion across the sediment-water interface, primarily along coastlines or the bottom boundary layer (Charette et al., 2015). Since $^{228}$Ra is more rapidly regenerated within sediments and its half-life is shorter than $^{226}$Ra, it is closely associated with coastal and continental shelf regions where its abundance decreases rapidly away from its source towards the open waters, thus tracking the sediment-ocean or shelf-ocean transition (van Beek et al., 2007; Burt et al., 2016b; Kadko and Muench,

2005; Kawakami and Kusakabe, 2008; Moore et al., 1980, 2008; Rutgers van der Loeff et al., 1995). For the purpose of this study, we assume that $^{228}$Ra has no pelagic source. Likewise, $^{226}$Ra is released from the sediment and disperses into the water column through advective and diffusive mixing (Charette et al., 2015; Grasshoff et al., 1999; IAEA, 2011), but given its long half-life, $^{226}$Ra can be distributed over great distances, often decaying within the oceanic water column (Charette et al., 2015;

IAEA, 1988). With the exception of a slight enrichment in Pacific Ocean deep waters relative to the deep waters of the Atlantic Ocean (Broecker et al., 1967; Charette et al., 2015; Chung, 1980), $^{226}$Ra reveals a nearly conservative distribution in the oceans facilitating its use as a long-term pelagic-based





tracer of water masses and of shelf inputs. These characteristics allow the two long-lived Ra isotopes to

be used as radioactive geochemical tracers to distinguish water-mass sources and their distribution

patterns, in our case within the CAA.

## II. Methods

### II.I. Sample Collection

During the summer of 2015 Canadian GEOTRACES cruise, 64 water samples were collected

throughout the Canadian Arctic Archipelago aboard the CCGS Amundsen at 17 different stations as a

subset of the overall biogeochemical sampling (Fig. 1). Samples for dissolved inorganic carbon (DIC),

total alkalinity (AT), barium (Ba), the stable oxygen isotope composition of water ($\delta^{18}O$), and Ra

isotopes were collected at various depths from the surface to 1000m on the up-cast of a rosette system

equipped with (24) 12-L Niskin bottles. Surface samples (2-12m) for Ra were collected using an

onboard pump collecting ship-side. In addition, temperature and salinity ($S_P$) measurements were

recorded on the downcast by a Sea-Bird SBE 9 (Seasave V 7.23.2) CTD throughout the water column.

The CTD salinity-probe measurements were calibrated post-cruise using a Guidline salinometer in the

home laboratory against taken discrete samples taken directly from the Niskin bottles into 250 mL

screw-cap HDPE bottles. DIC and AT samples were collected directly from the Niskin bottle into

250mL or 500mL borosilicate glass bottles to which 100µL of a saturated $HgCl_2$ solution was added

before being sealed with ground-glass stoppers, Apiezon® Type-M silicon grease and elastic closures

(Burt et al., 2016a). The bottles were then stored in the dark at room temperature or 4°C until they

could be processed on board. A VINDTA 3C (Versatile Instrument for the Determination of Titration

Alkalinity, Marianda) was used to firstly analyze the DIC samples by coulormetric titrations, and

secondly determine AT by potentiometric titrations (Shadwick et al., 2011a). A calibration of the

instrument was performed against certified reference materials (CRM) provided by Andrew Dickson





(Scripps Institution of Oceanography) and the reproducibility of the DIC and AT measurements was better than 0.1%.

Each Ra sample (105-215L) was sequentially pre-filtered through 10μm and 1μm filters, either directly using the ship's pump, or a high-volume pump connected to the Niskin bottles. The Ra isotopes were then preconcentrated by elution through manganese dioxide ($MnO_2$)-coated acrylic fiber cartridges at a constant flow rate of 1 L min$^{-1}$ (Charette et al., 2001; Grasshoff et al., 1999; Moore and Reid, 1973). To verify the extraction efficiency of the $MnO_2$ fiber cartridge, a second fiber-filter was occasionally mounted in series. $^{224}$Ra was then determined using a Radium Delayed Coincidence Counter (RaDeCC) (Moore, 2008), which had been calibrated against an IAEA (International Atomic Energy Agency) distributed reference material. The detection limit was estimated to 3 atoms L$^{-1}$ (0.05dpm 100L$^{-1}$) for $^{224}$Ra (see for details Moore, 2008; Moore and Arnold, 1996). No $^{224}$Ra activity could be detected in any of the second cartridges. Samples were then shipped to Woods Hole Oceanographic Institution to be ashed at 820°C for 16h, homogenized and transferred to counting vials (Charette et al., 2001). Well-type gamma spectrometers (Canberra and Ortec high purity germanium) were used to quantify $^{226}$Ra (via $^{214}$Pb @ 352 keV) and $^{228}$Ra (via $^{228}$Ac @ 338 and 911 keV) (IAEA, 2011). Each detector was calibrated with Mn-fiber ash spiked with a NIST-certified reference material #4967A ($^{226}$Ra) or a gravimetrically-prepared $Th(NO_3)_4$ solution with the $^{228}$Ra daughter in secular equilibrium with its parent $^{232}$Th. Detection limits, determined using the Currie Hypothesis test (De Geer, 2004), were determined to be 0.2 dpm for both $^{226}$Ra and $^{228}$Ra (Gonneea et al., 2013), which is equivalent to ~0.15 dpm 100 L$^{-1}$ for a typical 130 L sample.

Barium (Ba) concentrations were determined in water transferred directly from the Niskin bottles to 30mL HDPE plastic bottles containing 15μL of concentrated ultrapure hydrochloric acid (Thomas et al., 2011). Each subsample was then analyzed by Isotope Dilution using Sector Field Inductively Coupled Plasma Mass Spectrometry (SF-ICP-MS, Element 2, Thermo Finnigan) in Brussels. The





instrument was run in the low mass resolution mode m/Δm =300. Limit of detection and limit of

quantification based on blank analyses were: 0.06 and 0.20 nM, respectively (LOD = 3 X s.d. blank;

LOQ =10 X s.d. blank). Reproducibility of multiple measurements of reference materials (SLRS5;

SLRS3; OMP) was ≤ 2.5%. Details of the instrument's operational conditions are given by Thomas et

al. (2011). The barite saturation state ($Q_i$) is the ratio of the aqueous barium and sulfate ion activity

product ($Q_{(BaSO4, aq)}$) to the barite solubility product ($K_{Sp}$):

$$Saturation\ State\ BaSO_4(Q_i):\ = \frac{Q_{BaSO_4 aq}}{K_{Sp(Barite)}} \qquad \text{(eq. 1)}$$

As described in greater detail by Thomas et al.(2011), $Q_i$ has been computed after Monnin (1999) and

Hoppema et al. (2010).

Samples destined for measurements of the stable oxygen isotope composition of seawater ($\delta^{18}O$)

were taken directly from the Niskin bottles into 13mL screw cap plastic test tubes (Lansard et al.,

2012). The samples were analyzed at the GEOTOP-UQAM stable isotope laboratory using the $CO_2$

equilibrium method of Epstein and Mayeda (1953) on a Micromass Isoprime universal triple collector

isotope ratio mass spectrometer in dual inlet mode (Mucci et al., 2018). The data were normalized

against three internal reference waters, themselves calibrated against Vienna Standard Mean Ocean

Water (V-SMOW) and Vienna Standard Light Arctic Precipitation (V-SLAP). Data are reported on the

δ-scale in ‰ with respect to V-SMOW, and the average relative standard deviation on replicate

measurements is better than 0.05‰.

### II.II. Principal Component Analysis

The Principal Component Analyses (PCA) were performed to quantitatively determine the

correlation between variables as well as the affinity between each of the samples to arbitrary





components, while reducing the effects of random variation by using a correlation matrix (Gunasekaran

and Kasirajan, 2017; Peres-Neto et al., 2003). PCA is measured in Eigenvalues and Eigenvectors,

quantitative measures of the relative variation of variables along the axes as well as the loadings (or

coefficients) of each variable to the associated axes, respectively. Furthermore, PCA creates "best fit"

linear relationships between points within arbitrary space, providing a useful statistical tool to

distinguish associated trends, reduce dimensionality, and uncover relationships between related

variables within a three-dimensional space (Jolliffe and Cadima, 2016; Pearson, 1901). For this study,

associated or derived variables such as the radium isotopic ratios were excluded from the PCA due to

the congruency with other incorporated variables. Prior to statistical analysis, the variables from each

station and depth were transformed to fit a near-normal distribution and normalized to satisfy the

parameters of the analysis. Interpolations of the $\delta^{18}O$ and Ba data were made as samples were not

collected at every depth at each station. The interpolations were verified to the original data by means

of linear regression and comparison of slopes. Only three surface data samples were interpolated for Ba

samples, each from within Baffin Bay. After interpolation and normalization each sample was

categorized by depth: Surface, Middle Depth, Deep Archipelago, or Deep Atlantic, ranging from 0-

20m, 21-80m, 81-500m and >500m, respectively.

In addition, quantitative analyses of the PCA results were conducted by a broken stick analysis as a

means to distinguish the loading significance of each variable. To this end, eigenvectors were scaled to

V-vectors (product of eigenvector multiplied by the square root of the specific eigenvalue) and $V^2$

vectors (V-vectors$^2$) (Jackson, 2004; Peres-Neto et al., 2003). End members were calculated for each of

the variables significantly loading on PC1 from the derived partial values (Eigenvector for associated

variable multiplied by the PC score for that sample). Each partial value was then de-normalized and

back-transformed, thus deriving a refined rendition of the original data set (Appendix 1). Lastly, linear

regressions of each variable, with the exception of $^{228}Ra$, against the practical salinity were plotted to





express robust end-member relationships from within the previously categorized salinity-defined water

masses present throughout the CAA. We report the respective end-members as "apparent end-members", as they resemble the mean end-member properties in the CAA. For example, an apparent freshwater end-member at null practical salinity ($S_P$=0) would be composed of various individual river and meteoric water end-members, in consideration of their relative weights in this composite. As $^{228}$Ra originates from sediments, which in our study are primarily located in waters at the lower salinity range

of the CAA, the extrapolation was done to $S_P$ = 25 (eg. Rutgers van der Loeff et al., 2003).

### III. Results and Discussion

#### III.I. Water Mass Properties

Surface values of practical salinity ($S_p$), density, DIC, AT, and δ$^{18}$O were found to increase from

west to east through the CAA (Figs. 1, 2a, c, d, f, g, Appendix 2). This trend was extended to the temperature profiles taken throughout the CAA, with the exception of station CAA5, which was found to closely resemble the temperature profile of CAA3 (Fig. 2b, Appendix 2). Prinsenberg and Bennett (1987) reported similar results from samples collected in 1982 across Barrow Strait, a sill less than 200m deep located roughly between 105°W to 90°W, where analogous transects for salinity and

temperature were recorded throughout the surface layer (Fig. 1). This is both the widest and shallowest section of the CAA. It is responsible for restricting the eastward flow of Deep ATL waters found in the Western Canadian Basin and inhibiting high salinity ($S_p$ > 33.1) ATL water within Baffin Bay from venturing westward (Hamilton and Wu, 2013; Jones, 2003; Prinsenberg, 1982; Prinsenberg and Bennett, 1987; Shadwick et al., 2011a; Yamamoto-Kawai et al., 2010).

Contrasting these trends, dissolved Ba, $^{226}$Ra and $^{228}$Ra concentrations have been found to decrease eastward both at the surface as well as at the mid-depth maximum, which is a result of the elevated flow rates, increasing distance from their source within the CAA, and proximity to the Ba- and $^{228}$Ra-





depleted ATL waters in Baffin Bay (Figs. 2e, 3, Appendix 2 and 3) (Thomas et al., 2011). In general, the $^{228}$Ra/$^{226}$Ra isotopic ratio (Fig. 3c) decreases with depth, but occasionally follows a more complex

spatial pattern, which will be unraveled at the end of this paper.

The property-property diagrams of DIC and AT vs. $S_p$ display strong positive relationships in surface waters (with $S_p$=0 intercepts of 371 µmol DIC kg$^{-1}$ and 492 µmol AT kg$^{-1}$, respectively). This implies the addition of freshwater by means of sea-ice melt (SIM) and Meteoric Water (MW), thus attributed to the PML found in the surface waters of the stations west of 96.5°W (Figs. 1, 2, 4, 5c). Whereas both

DIC and AT concentrations increase along the surface from west to east throughout the CAA, the highest DIC concentrations were observed at the pycnocline of the western most station (CB4) (Fig. 2 and Fig. 5a). This maximum in metabolic (respiratory) DIC decreases slightly eastward due to the increasing contribution of low-DIC ATL waters (Fig. 5a) (Shadwick et al., 2011a). Similar results were observed for AT, although AT concentrations were also found to increase both with depth (Figs. 2d, g,

4). This is explained by the concomitant increase in AT and $S_p$ values rather than metabolic activity, thus distinguishing AT from DIC (Burt et al., 2016a; Shadwick et al., 2011a; Thomas et al., 2011).

Despite the intrusion of deep Atlantic Ocean waters throughout the CAA (Jones, 2003; Newton and Coachman, 1974), CB4 (in the Canada Basin) displays different ATL, UHL and PML water-mass characteristics than those observed at stations within the CAA (Appendix 2 and 3). Hence, for the

remainder of this study, we will ignore data from CB4 in our discussion of the circulation in the CAA, although we will return to the role and positioning of CB4 in relation to the CAA waters at a later stage of the paper, particularly in relation to the $^{228}$Ra/$^{226}$Ra ratio.

In order to identify water-mass distributions and mixing regimes within the CAA, DIC was normalized to a constant salinity (DIC$_{norm}$) (eq. 2). DIC$_{norm}$ is a powerful tool to gain a better

understanding of water mass characteristics and distributions by eliminating the influence of fresh water inputs and thus highlighting alternative non-conservative controlling parameters such as





biological processes at the time scale of mixing (Friis et al., 2003; Shadwick et al., 2011b).

$$DIC_{norm} = ((DIC_{measured} - DIC_{S=0}) * S_{measured}^{-1}) * S_{reference} + DIC_{S=0}$$ (eq. 2)


The distribution of $DIC_{norm}$ displays an eastward decrease in surface waters along the eastward bulk flow throughout the CAA, consistent with observations that surface DIC values were lowest in the Western samples (Figs. 1, 2g, h, 5b, c). The reversal in trend reflects the decrease in accumulated respiratory DIC as waters flow longitudinally eastward through the CAA. The presence of two distinct, non-mixing water-masses were highlighted through the $DIC_{norm}$– $S_p$ relationship, distinguishing the surface (PML) from subsurface samples (Figs. 1 and 5c). These results confirm the previous distinction proposed, based on the distribution of Ba, between waters above and below the UHL across the pycnocline (Figs. 2, 5b, c) (Shadwick et al., 2011a; Thomas et al., 2011).


III.II. The use of Radium Isotopes as Water Mass Tracers


Radium isotopes, specifically [226]Ra, [228]Ra and their ratio ([228]Ra/[226]Ra), were used as proxies to reconstruct the water mass distribution throughout the CAA. Like the chemical constituents DIC, AT, $\delta^{18}O$ and Ba, as well as the stable oxygen isotopic composition of waters, Ra isotope activity and ratios were found to vary between stations, with depth, as well as across water masses (Appendix 3). The highest [228]Ra activities were observed at the surface, particularly at the shallow stations 312 and 314, located in the center of the CAA, and thus more strongly influenced by sedimentary [228]Ra release (Figs. 1, 6). Lower [228]Ra activities were found at higher salinities and depths, comparable to values reported by Burt et al. (2016b) in the North Sea, where elevated [228]Ra activities were present within the shallow, lower salinity waters. Like the $DIC_{norm}$- $S_p$ relationship, the [228]Ra- $S_p$ relationship (Figs. 5b, c, 6) reveals two distinct water masses that separate the surface (PML) from subsurface waters. For the surface



sample grouping, a negative slope ($^{228}$Ra-S$_P$) was obtained (slope= -2.467x), whereas for the deeper

samples a less negative slope was found (slope= -0.4854x). The more negative slope, associated with

the surface samples collected throughout the CAA, indicates that the system is heavily influenced by

the influx of $^{228}$Ra from the CAA shelf sediments. (Fig. 6). In contrast, the slope derived from the $^{228}$Ra

activities recorded in the deeper waters of the CAA may imply that these samples originate from an

open ocean setting, with minimal (or much less) contact with continental shelf or coastline sediments

over the past few decades. As noted earlier (Fig. 3, 6b), the $^{228}$Ra activities decrease from west to east

through the CAA in both the surface and deep samples. These values are interpreted as reflecting the

mixing of Pacific waters with Atlantic (Baffin Bay) waters east of Barrow Strait. We provide a more

detailed analysis of the $^{228}$Ra activity distribution pattern below (section III.III.I.II).

### III.III  Characterizing Water Masses and Isolating End-Members through Principal Component Analysis (PCA).

Further investigation of the dominant water-mass patterns was undertaken through Principal

Component Analyses (PCA). The first and second principal components (PCs) accounted for 59.1%

and 17.5% (total 76.6%), respectively, of the variability in the data (Fig. 7, Table 1). The third PC

accounted for a further 13.2% of the variability (89.8% in total). The fourth and fifth PCs together

accounted for less than 10% of the variability and were not included in subsequent analyses. PC1, in

turn was inverted to establish apparent end-members of the source waters found in the CAA (section

III.III.II.I, see methods II.II).

### III.III.I Qualitative Analysis of PCA

### III.III.I.I Surface Water Mass Distinction

The first PC (PC1) loaded very heavily on salinity, AT, and δ$^{18}$O, accounting for 94–97% of the




variability in each (see Table 1). It also loaded heavily on DIC and $^{228}$Ra (67% and 66% of variability)
and less heavily on Ba (37%). The latter five parameters were all inversely correlated with salinity (Fig.
7). The second PC (PC2) loaded heavily on temperature (83% of variability) and relatively weakly on
DIC (a further 22% of variability for a total of 89% between PC1 and PC2) and Ba (34%, for a total of
71% between PC1 and PC2). Temperature and DIC were directly correlated (Fig. 7), so the component
of DIC accounted for by PC2 cannot be ascribed to temperature-dependent solubility. The third PC
(PC3), which only accounted for 13.2% of the variability in the data, loaded on $^{226}$Ra (74% of its
variability) and was the only PC that did so.

The ordination of samples on PC1 and PC2 shows a strong separation between surface and mid-
depth samples vs deep samples (Fig. 7), reflecting the consistent differences in their parameter values
(Figs. 2-6). Variability within surface and subsurface layers was examined by re-running the analysis,
using only these data, to minimize the influence by the systematically-different deep-water data on their
ordination. The restricted analysis retained most of the parameter relationships observed in the full
PCA (Fig. 7, 8). The first PC explained slightly more of the variability (63.5% vs 59.15%), while the
second PC explained slight less (14.7% vs 17.5%), for a total of 77.2% vs 76.7%. There was strong
loading of salinity, AT, $\delta^{18}$O, DIC and $^{228}$Ra on PC1, with the latter four inversely correlated with
salinity. In contrast, though, temperature was strongly correlated with salinity rather than orthogonal to
it and Ba was strongly loaded on PC2.

The re-ordination of the surface and mid-depth data indicates a strong geographic separation of the
samples on PC1 (Fig. 8), which is also evidenced by temperature-salinity and $^{228}$Ra/$^{226}$Ra-DIC plots
(Fig. 9). The first surface group comprises samples from the eastern edge of the CAA, under the
influence of Atlantic waters, which enter the CAA via Baffin Bay and Lancaster Sound. The second
group comprises the PML-influenced surface samples from the two southern interior CAA stations
(312 and 314) and the northwestern CAA stations (CAA4-CAA7), and the mid-water samples (also





from Stations 312 and 314). It is worth noting that the outer-most surface west samples, likely best

visible in Fig. 9b, are sampled from station CB4. This attribution will be explained later with reference

to the apparent end-member properties.

III.III.I.II Distinction of Deep-Water Masses and Indication of Flow

The ordination in the initial PCA is analogous to a T-S diagram, given that PC1 loads on salinity and

its covariates and PC2 loads most heavily on temperature. There are very strong similarities between

deep-water samples collected in Baffin Bay (Fig. 7; Deep ATL). The deep-water samples within the

CAA are ordinated along a gradient between the Deep ATL samples and an end-member that would

have negative factor loadings on PC1 and PC2. This would likely be Pacific water. The two deep

samples from the westernmost station, CB4, are anomalous (Fig. 7). Their ordination suggests that they

are an end-member for the Deep ATL water, this is due to Deep ATL waters that flow west past

Svalbard, before crossing the Lomonosov Ridge and accumulating in the Canada Basin (Coachman and

Barnes, 1963; Newton and Coachman, 1974).

Paradoxically, samples collected at Station CAA3 are found at both ends of this trend, having both

the highest and lowest similarity to the Deep ATL samples of samples within the CAA. The three deep-

water samples collected at CAA5 are intermediate between the Deep ATL and deep CAA3 waters

compared to the remainder of the CAA. The very strong similarity between the deep-water samples at

CAA3 and Baffin Bay indicates that they are ATL water that recirculated counter-clockwise around

Baffin Bay and combined with Arctic outflow through Nares Strait  (Bâcle et al., 2002; Curry et al.,

2011; Lobb et al., 2003). A third PCA was performed excluding the alkalinity (which clearly expresses

the dominant bulk eastward flow), to visualize the transition of Deep ATL water as it mixes with the

UHL in the CAA (Fig. 10). Results of this analysis reveal that the Deep Arch stations CAA1, CAA3

and CAA5 are more closely linked to the Deep ATL group, implying that they are in fact part of the





Deep ATL water mass (Fig. 10). This suggests that there is an intrusion of ATL water along the northern

edge of the CAA. This westward flow with a speed of 2.2 cm/s was observed by Prinsenberg and

colleagues (2009) and is weaker than the dominant eastward current flow (15.3 cm/s).

There is support from observations of dissolved Ba and the barite saturation states ($Q_i$) along the

north-to-south transect across the eastern Lancaster Sound (CAA1, CAA2, 323, 324, CAA3) (Fig. 11a,

b). An increase in Ba and $Q_i$ was observed from north to south at the surface as well as at depth. Lower

Ba concentrations and barite saturation states have been observed in Baffin Bay waters, which are fed

by the West Greenland Current. In contrast, the continentally-impacted waters from the CAA are

characterized by significantly higher values for the two Ba properties, such that the origin of the waters

from the Atlantic or the CAA can be discriminated clearly (Thomas et al., 2011). Furthermore,

substantially lower values of the $^{228}Ra/^{226}Ra$ ratio (Fig. 11c) indicate the inflow of Atlantic water on the

northern side of Lancaster Sound as well as its outflow along its southern side. The same pattern is

revealed by $^{226}Ra/Ba$ (Fig. 11d), which is dominated by the $^{226}Ra$ variability (Fig 13b). The lowest $^{226}Ra/$

Ba value reflects the inflow of low $^{226}Ra$ waters from the Atlantic Ocean (Fig. 11d), as the observed

$^{226}Ra$ activities (8-9 dpm 100L$^{-1}$, Fig. 3a) are in the same range as those measured in the surface waters

of the Atlantic Ocean (Le Roy et al., 2018), part of which feed into the West Greenland current. Since

$^{226}Ra$ activities reveal a much larger north-to-south gradient across Lancaster Sound than Ba does (Figs.

2e, 3a), the seemingly opposing gradients shown in Figs. 11c and 11d are dominated by changes in

$^{226}Ra$. The waters transiting through the CAA and exiting on its southern side are enriched with both

$^{228}Ra$ and $^{226}Ra$ as they interact with the shallow sediments. Differences in the two isotopes' half-lives

underlie the gradient in the ratio (Figs. 11c, 12). The Ba and Ra data are consistent in indicating bi-

directional flow linking the northern and southern stations along 80°W. This can be attributed to the

counter-clockwise, cyclonic circulation found throughout Baffin Bay (Figs. 11 and 13).

A closer look at the $^{226}Ra$-Ba relationship reveals the association between water-masses observed in




the CAA and those of the surrounding oceans, in particular to the Atlantic Ocean. The highest surface

values of the [226]Ra/Ba ratio are observed in the interior of the CAA, while the lowest are found in the

Canada Basin and the eastern side of the CAA (Fig. 13d). The direct relationship between [226]Ra

activities and Ba concentrations shows that the Arctic samples with a $S_P > 34$, i.e., waters of Atlantic

origin, fall along the relationship established by Le Roy et al. (2018) for the Atlantic, or reported by

van Beek et al. (2007) for the Sargasso Sea. This relationship, in turn, is similar to the one for the world

ocean established from the GEOSECS data-base (Le Roy et al., 2018). This implies that the deep

Lancaster Sound samples, as well as the deep Canada Basin samples, can similarly be linked to an

Atlantic origin. The remaining CAA samples, with a $S_P < 34$ display a clear deviation from this

relationship and towards higher Ba values (Fig. 13b), which can be attributed to the high Ba runoff

from rivers draining into the CAA. The open-ocean [226]Ra/Ba ratio has been reported to be relatively

constant at about 2.2-2.5 dpm ($10^{-6}$mol$^{-1}$), with elevated values observed only near deep-ocean

sediments (van Beek et al., 2007; Le Roy et al., 2018). In contrast, the CAA data show a wider range of

[226]Ra/Ba values, which appear to be strongly controlled by the [226]Ra activity, rather than variability of

the Ba concentration (Fig. 13). The [226]Ra/Ba ratio in water masses of Atlantic origin ($S_P > 34$) are offset

toward higher ratios for a given [226]Ra activity, a consequence of the relatively higher Ba content of the

water masses transiting through the CAA. The highest [226]Ra/Ba surface values were observed in the

interior of the CAA, whereas the lowest ones were measured in the Canada Basin and the eastern side

of the CAA (Fig. 13d).

III.III.II Interpretations of PC1 and PC2

III.III.II.I: Principal Component One: Advection / Land-Ocean Transition

PC1 was found to correlate significantly with $S_p$, DIC, AT, $\delta^{18}$O, [228]Ra and Ba, suggesting that this

axis represents the land-ocean gradient, i.e. the advective (estuarine) mixing regime of fresh and salt





water (Table 1, Fig. 7). This interpretation is consistent with our previous attribution of the longitudinal, eastward increase in $S_p$, DIC, and AT of surface waters through the CAA to a decreasing coastal influence (Figs. 3, 4, 5).

Here, the [228]Ra activity has to be viewed from a somewhat different perspective, as the

sedimentary/shelf sources also reside in the low salinity range of our samples ($S_p \sim$ 25-30), but does not align with any riverine source (e.g., Rutgers van der Loeff et al., 2003, see below). Therefore, in regards to the PC1 axis, [228]Ra relative to [226]Ra, represents the coastal, shelf to open ocean transition, decreasing in activity laterally as waters primarily follow the bulk eastward flow and are transported away from the sedimentary source within the CAA (Figs. 12, 13, see also Charette et al., 2016).

Accordingly, the end-member was defined with $S_P$ = 25.

The loading of PC1 on $\delta^{18}$O and Ba is also high, particularly for $\delta^{18}$O. These variables allow us to discriminate between freshwater sources (MW vs. SIM) while also demonstrating a clear mixing gradient along the land-ocean transition (Guay et al., 2009; Macdonald et al., 1999; Yamamoto-Kawai et al., 2010).

We exploit the relationships of salinity to the individual properties derived from PC1 in order to define the freshwater and marine (saline) end-members (Table 2). Again, these end-members should be considered as "apparent" end-members, as these represent observed mean end-member properties, and not end-member water masses, for example, end-members of individual rivers. "Apparent" end-members for each of the significant loading variables associated with PC1 were calculated (Table 2). It

should be noted that [226]Ra was included in the PC1 analysis even though it primarily weighs on the PC3 axis (73.8%) not the PC1 axis. This is because salinity is not a significant loading factor on the PC3 axis (Table 1). Therefore, with the exception of PC3, [226]Ra next most closely associates with the PC1 axis (18.6%), thus allowing for PC1 to be used to establish the [226]Ra "mixing" endmember (Table 1).

The computed "apparent" end-members sit along the mixing curve, many located "halfway"





between the SIM and MW values reported in the literature (e.g., Cooper et al., 2005; Guay and Falkner, 1998; Macdonald et al., 1989; Shadwick et al., 2011a; Thomas et al., 2011). These "apparent" freshwater end-members ($S_P$=0) thus mirror the combination of freshwater sources. Since both the MW and SIM are equally represented in the "apparent" DIC, AT and Ba end-members, it can be assumed that the freshwater end-member is located within the western portion of the CAA, as the freshwater

contribution to the eastern ATL water mass is dominated by SIM, with little to no MW (Shadwick et al., 2011a, 2011b). The "apparent" end-member surface values for $\delta^{18}O$ were found to closely resemble MW (rather than the larger SIM values), with MW being, in essence, the dominant source of freshwater (Table 2) (e.g., Thomas et al., 2011). The end-members associated with the UHL and ATL were also found to be very similar to those reported in the literature, especially AT, Ba, and $\delta^{18}O$ (Guay et al.,

1997; Guay and Falkner, 1998; Macdonald et al., 1989; Shadwick et al., 2011a; Thomas et al., 2011; Yamamoto-Kawai et al., 2010). The "apparent" DIC end-members for these water masses do not closely concur with literature values, nor do those produced here (Fig. 5a), as the DIC end-member associated with the UHL is expected to be larger than that of the ATL (e.g., Shadwick et al., 2011a). We argue that this is due to the impact of biological processes, which cannot be resolved solely from

mixing-conservative properties. These characteristics will be discussed within PC2. Furthermore, this result may be due to the normalization required for the PCA linearization of the Deep ATL and Arch samples, thus diminishing the characteristic DIC maximum at the pycnocline. Lastly, we propose apparent end-members for $^{226}Ra$ and $^{228}Ra$ in the CAA. The highest $^{228}Ra$ and $^{226}Ra$ activities were found in the fresh(er) sources attributed to the surface samples collected in the western CAA (Figure 6). The

$S_P = 0$ end-member for $^{226}Ra$ (25.2 dpm 100L$^{-1}$) is consistent with the effective $^{226}Ra$ end-member for the Mackenzie River (26.1 dpm 100L$^{-1}$; Kipp et al., 2019), which contributes to the freshwater budget of the CAA. Smith et al. (2003) reported a Beaufort shelf end-member with a $^{228}Ra$ activity of 12 dpm 100L$^{-1}$ at $S_P$= 2 and a $^{228}Ra/^{226}Ra$ of ~1. Therefore, with a shelf apparent endmember for $^{228}Ra$ of 22.4



dpm 100L$^{-1}$ at $S_P$= 25 (Table 2) and a $^{228}$Ra/$^{226}$Ra of ~2, the shelf sediment influence of $^{228}$Ra in the CAA
is quite conclusive.

Similar apparent end-member $^{226}$Ra activities were observed within the open waters of the UHL and
ATL, while substantially lower $^{228}$Ra activities were recorded in the ATL (Table 2). Rutgers van der
Loeff et al. (2003) reported a high-salinity $^{226}$Ra end-members in a similar range (~6-9 dpm 100L$^{-1}$) to
the apparent ones reported here, while the apparent $^{228}$Ra end-members obtained here are clearly lower
than those reported by Rutgers van der Loeff et al. (2003, ~3.2-15.4 dpm 100L$^{-1}$). An obvious
explanation for this discrepancy may be the circulation history of the respective water masses, as the
Atlantic end-member in the CAA most likely has a longer circulation history than on the Eurasian side.
Furthermore, the salinity of the samples reported in this study are higher than the samples measured by
Rutgers van der Loeff et al. (2003), implying the presence of a stronger ATL component in our samples.
The differences between the high salinity, apparent, $^{226}$Ra and $^{228}$Ra end-members, might reflect their
vastly different half-lives, allowing for a tangible decay of $^{228}$Ra at oceanic transport timescales in
contrast to the "nearly conservative" $^{226}$Ra. Coinciding with the previous result, higher variability in
$^{228}$Ra was seen throughout the water column, where $^{226}$Ra activities varied only slightly. Overall, the
identification of Ra end-members in the region highlights the Ra sources and transport pathways
throughout this complex coastal/shelf environment.

We use the derived apparent end-member properties of $^{228}$Ra and $^{226}$Ra to gain further insight into the
distributions of the two isotopes and thus the flow pattern within the CAA. The $^{228}$Ra/$^{226}$Ra was
computed as a function of salinity and compared to the relationship between the $^{228}$Ra/$^{226}$Ra and $\delta^{18}$O
(Fig. 12a, b). The apparent $^{228}$Ra/$^{226}$Ra over $S_P$ mixing ratio appears as if the ratio was only affected by
conservative mixing of the two respective end-members (Table 2). When relating this ideal behaviour
to the ratios observed in our study, three main groups of samples can be identified. A: the higher
salinity ($S_P$>32, $\delta^{18}$O>~-3) samples, that more or less fall together with the mixing relationship. B: the





samples characterized by substantially higher $^{228}$Ra/$^{226}$Ra (~27<$S_P$<30, ~-5<$\delta^{18}$O<-3), and C: the second

group of low-salinity samples with ~$S_P$<31 ($\delta^{18}$O<~-3), characterized by a substantially lower

$^{228}$Ra/$^{226}$Ra isotopic ratio. The spatial distribution (Fig. 12c) of these sample groups unravels processes

that shape the Ra distributions, that, at the first view, did not seem to fit into the broader scheme

described in Fig. 9. Samples with higher $^{228}$Ra:$^{226}$Ra than the mixing ratios are located within the CAA,

at stations 312 and 314, and at the downstream stations along the southern coast of the northwest

passage, which in turn are under the strong influence of shelf/sediment derived $^{228}$Ra accumulation as

they flow eastward. This water mass mixes on the southern side of Lancaster Sound with the water

from Baffin Bay, yielding a flow pattern highlighted by the higher $^{228}$Ra/$^{226}$Ra ratios in the CAA (Fig.

11c, 12c). The stations with lower Ra isotopic ratios than the mixing ratios are located on the northern

part of the Lancaster Sound and connect to the Canada Basin via McClure Strait and Parry Channel

(Fig. 1, 12c). The waters at stations with lower Ra isotopic ratios mix with (inflowing) water from

Baffin Bay along the northern side of Lancaster Sound (Fig. 11c). The overall lower $^{228}$Ra/$^{226}$Ra waters

reflect the long-term isolation of CB4 waters from their margin source (e.g., Kipp et al., 2018) such that

the $^{228}$Ra activities are diminished noticeably by radioactive decay. Consistent with this finding is the

clear separation between water of the northern and southern sides of Lancaster Sound, as discussed in

Fig. 11. We integrate the observations and findings featured in Figs. 11-13 into a revised scheme,

shown in Fig. 14, to reveal the main flow pattern.

This analysis can further be exploited to highlight the release of $^{228}$Ra from shallow shelf sediments

to waters in the lower salinity range rather than from rivers. Both the $^{228}$Ra/$^{226}$Ra- $S_P$ and $\delta^{18}$O reveal a

non-conservative addition of $^{228}$Ra to waters in the salinity range of 25<$S_P$<30 (Fig. 12). Furthermore,

when considering $\delta^{18}$O, conservative mixing from a riverine source can be excluded (see also Burt et

al., 2016b; Kipp et al., 2018; Moore, 2000, 2007). On the other hand, the $\delta^{18}$O values of -3 - -4 ‰

imply that the $^{228}$Ra source is under riverine influence, as the $\delta^{18}$O signature of the sea-ice end-member





is generally thought to be approximately -2‰ (e.g., Eicken et al., 2002; Thomas et al., 2011; Yamamoto-Kawai et al., 2009, and references therein, see also Thomas et al., 2011 Fig. 5d).

III.III.II.II: Principal Component Two: Particle-related impacts (nutrient-type behaviour)

The correlation of Ba, DIC and temperature with PC2 is based on the hydrographic peculiarities of the CAA, where temperature displays a "classical", inverse nutrient-type profile (Fig. 2), resulting from the presence of a temperature minimum in the UHL. As is the case for Ba and DIC, nutrient-type profiles are generally shaped by the interaction of biological (production/respiration, 515 adsorption/desorption) processes and gravitational particle settling. Properties revealing such distributions are represented by PC2, with the temperature minimum coinciding with those minima found at the pycnoclines depth.

**IV. Conclusions**

It is our hope that with a better understanding of the distributions of the long-lived Ra isotopes, coupled with other chemical constituents, future initiatives can be supported to investigate the impacts of climate change within this region. Given the results of the PCA as well as the presence of $^{228}$Ra, our data are consistent with the existence of a western flow of water along the Northeastern edge of the CAA. This flow pattern coincides with the U-turn of cyclonic Baffin Bay originating water, intruding 525 westward into the CAA before being rerouting back to the east. The bulk eastward transport of water through the CAA was confirmed, highlighted by the distribution of Ra radioisotopes and chemical constituents in end-members throughout the region. Overall, the results from this study provide the foundation for future GEOTRACES studies or other initiatives that focus on the sensitivity of trace element fluxes to changing environmental conditions by identifying and quantifying anomalies in the 530 distribution of radioactive isotopes in the Canadian Arctic Archipelago. Furthermore, this study

provides an additional tool to better understand the vulnerability of ecozones, such as the Arctic to climate change by characterizing water mass distributions, flow patterns, mixing and their respective time scales in these challenging sampling areas.

**Acknowledgments:**

We wish to thank the captains and crew of the icebreaker CCGS Amundsen as well as the chief scientist, Roger Francois and his team, for their support at sea. We would also like to extend our appreciation to Jacoba Mol and colleagues on the ship for their collaboration. This study was supported by the Canadian GEOTRACES program, as part of the NSERC-CCAR initiative. MAC and PBH were

supported by U.S. GEOTRACES via NSF Chemical Oceanography program (#OCE-1458305). FD is grateful to J. Navez, M. Leermakers and K.H. Niroshana for assistance during the Ba analyses in Brussels. HTH acknowledges support by the German Academic Exchange service (DAAD, MOPGA-GRI, #57429828) supported by funds of the German Federal Ministry of Education and Research (BMBF).

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



**Figure Captions:**

**Figure 1:** Map of the Canadian Arctic Archipelago showing 17 stations sampled during the 2015 GEOTRACES cruise aboard the CCGS Amundsen (red dots), where the two unlabeled stations along the Eastern CAA cross channel transect are the surface stations 323 and 324, numbers refer to CAA stations (1-7), Nares Strait (NS), Barrow Strait (BS), McClure Strait (MS), Lancaster Sound (LS) and Parry Channel (PC) are denoted, the latter one connecting the CAA from MS via BS to LS. Lastly, the blue and grey lines indicate the 200m and 100m isobaths, respectively.

**Figure 2:** Vertical distributions of salinity, temperature, density, AT, Ba, $\delta^{18}O$, DIC and normalized DIC (DIC$_{norm}$) observed at 7 stations throughout the CAA.

**Figure 3:** $^{226}Ra$ (a), $^{228}Ra$ (b) and $^{228}Ra/^{226}Ra$ (c) profiles ranging from surface to depth for stations CAA1-CAA7 (where pink dots indicate station CAA2 which was only sampled at the surface and bottom) throughout the Canadian Arctic Archipelago.

**Figure 4**: Total Alkalinity (AT) versus salinity measured throughout the CAA with coloured depth (a) and longitude (b).

**Figure 5:** Dissolved Inorganic Carbon (DIC) (a) and salinity-normalized Dissolved Inorganic Carbon (DIC$_{norm}$) (b) were plotted against salinity ($S_p$), with colours indicating depth (m) (a, b) and longitude (c), the DIC$_{norm}$ vs. $S_p$ plots were fitted with a piecewise regression analysis representing the surface, y=-19.661x+2680.9 ($R^2$=0.533) and at depth; y=-34.186x+3282.2 ($R^2$=0.765). In plots (b & c) CB4 was excluded from the piecewise regression (represented by unfilled, crossed out grey circles), while stations 312 and 314 surface samples were excluded entirely. The black diamonds identify the average Atlantic deep-water samples from stations CAA1, CAA2 and CAA3.





**Figure 6:** $^{228}$Ra (dpm/100L) plotted against practical salinity with colour indicating depth (a) and longitude (b) fitted with a piecewise regression excluding the deep stations of the Canada Basin (grey diamonds) and yielding y = -2.4666x + 86.377 ($R^2$ = 0.2835) for the surface trend and y = -0.4854x + 21.127 ($R^2$ = 0.0722) for the deep trend. The average Atlantic deep waters sampled from stations CAA1, CAA2 and CAA3 is defined by a black diamond.

**Figure 7:** Eight-variable Principal Component Analysis (PCA) of PC1 and PC2 for 64 samples from 17 stations throughout the Canadian Arctic Archipelago, distinguished by depth groupings; Surface (0-20m; purple), Mid (20-80m; blue), Deep Archipelago (Deep Arch, 80-500m; red) and Deep Atlantic (Deep ATL, >500m; green). The sample collected at station CB4 at 200m depth was excluded from this plot. The ellipses represent 95% confidence intervals associated with each water mass grouping.

**Figure 8:** Eight-variable Principal Component Analysis of surface samples (0-20m;) east (green) and west (blue) and mid-Depth (20-80m; red) samples collected throughout the Canadian Arctic Archipelago analyzing PC1 and PC2 for 27 samples from 17 stations, with the exception of the surface sample collected at station CB4 which was excluded from this plot. The ellipses represent 95% confidence intervals associated with each water mass grouping.

**Figure 9:** Temperature-Salinity plots with colours indicating depth (a) and longitude (b) as well as the Radium Isotopic Ratio($^{228}$Ra/$^{226}$Ra)-DIC plots with colour indicating depth (c) and Salinity (d), furthermore highlighting three water masses throughout the CAA, the two surface water masses the Western Surface (Surf W) and Eastern Surface (Surf E) waters and one at depth (Deep).

**Figure 10:** Principal Component Analysis (PCA) of PC1 and PC2 for 64 samples from 17 stations throughout the Canadian Arctic Archipelago, composed of the seven normalized variables Salinity (S), Temperature (T), DIC, Ba, $\delta^{18}$O, $^{228}$Ra, and $^{226}$Ra, excluding AT. They are distinguished by depth groupings; Surface (0-20m; purple), Mid (20-80m; blue), Deep Archipelago (Deep Arch, 80-500m; red)

and Deep Atlantic (Deep ATL, >500m; green).

**Figure 11:** Cross section at stations CAA1-3, 323 and 324 for dissolved Ba (a) and barite saturation state ($Q_i$) (b), as well as $^{228}Ra/^{226}Ra$ (c) and $^{226}Ra/Ba$ (dpm µmol$^{-1}$) (d). The low values of both properties indicate the presence of Atlantic water (see Thomas et al., 2011).


**Figure 12:** Relationship between the $^{228}Ra/^{226}Ra$ as derived from the apparent endmembers vs. salinity (a) and $\delta^{18}O$ (b). $^{228}Ra/^{226}Ra$ in surface samples across the CAA depicting the different flow pass via the CAA/northwest passage, and via McClure Strait, Parry Channel and Lancaster Sound (c), black, white and gray stars group samples into water masses with high salinity and low isotopic ratio, with low

salinity and high isotopic ratio, and lastly with low salinity and low isotopic ratio, respectively.

**Figure 13:** Relationships between $^{226}Ra$ and Ba (a), and between the $^{226}Ra/Ba$ and $^{226}Ra$ and Ba concentrations (b), respectively (c). The red symbols indicate samples with $S_P>34$ (Atlantic origin). In (b) the linear regressions yield for samples with $S_P>34$ f(x) = 0.20x + 0.25, $R^2$=0.99, and $S_P<34$ f(x) =

0.18x + 0.05, $R^2$=0.90. Panel (d) depicts the spatial distribution of the $^{226}Ra/Ba$ in surface waters across the CAA.

**Figure 14:** Sketch of proposed flow pattern as identified in the current study.






**Appendices:**

**Appendix 1:** Equations used to normalize ($X_0$) the data distribution for Temperature, Salinity, Dissolved Inorganic Carbon (DIC), $\delta^{18}O$, $^{226}Ra$, $^{228}Ra$ and Ba collected throughout the CAA 2015 GEOTRACES cruise in preparation for Principal Component Analyses.


**Appendix 2:** DIC, AT, $\delta^{18}O$, and Ba plotted against depth for each station including CAA1, CAA2, CAA3, CAA4, CAA5. CAA6, CB4, 312, and 314, where profiles were taken throughout the 2015 GEOTRACES cruise in the Canadian Arctic Archipelago. Colours indicate the water masses present at the sampled depth; red is the Polar Mixed Layer (PML), yellow is the Upper Halocline Layer (UHL),

and blue is the Atlantic Layer (ATL).

**Appendix 3:** $^{226}Ra$, $^{228}Ra$, and Ra isotopic ratio ($^{228}Ra/^{226}Ra$) plotted against depth for each station, including CAA1, CAA2, CAA3, CAA4, CAA5, CAA6, and CAA7, where profiles were taken throughout the 2015 GEOTRACES cruise in the Canadian Arctic Archipelago. Colours of depth

indicate water masses at the sampled depths; red is the Polar Mixed Layer (PML), yellow is the Upper Halocline Layer (UHL), and blue is the Atlantic Layer (ATL).





**Table 1** Eigenvalues and Normalized $V^2$ Vectors for Temperature, Salinity, Dissolved Inorganic Carbon (DIC), Total Alkalinity (AT), $\delta^{18}O$, $^{226}Ra$, $^{228}Ra$ and Barium (Ba), where bolded values represent significant weight attributed to that Principal Component (where PC 4 and 5 were not analyzed, shaded) derived from the original 8-variable PCA (Fig. 7).

| PC | Eigenvalues | Temperature | Salinity | DIC | AT | $\delta^{18}O$ | $^{226}Ra$ | $^{228}Ra$ | Ba |
|---|---|---|---|---|---|---|---|---|---|
| 1 | 4.74 | 0.007 | **0.968** | **0.666** | **0.947** | **0.939** | 0.186 | **0.656** | **0.372** |
| 2 | 1.41 | **0.834** | 0 | **0.224** | 0.008 | 0.003 | 0 | 0 | **0.34** |
| 3 | 1.03 | 0 | 0.022 | 0.046 | 0.006 | 0.038 | **0.738** | 0.154 | 0.027 |
| 4 | 0.486 | 0.135 | 0 | 0.004 | 0.009 | 0 | 0.03 | 0.059 | **0.248** |
| 5 | 0.246 | 0.024 | 0.001 | 0.038 | 0.002 | 0 | 0.043 | 0.128 | 0.009 |

**Table 2** Apparent Dissolved Inorganic Carbon (DIC), Total Alkalinity (AT), $\delta^{18}O$, Barium (Ba), $^{226}Ra$, and $^{228}Ra$ end members, analyzed for the salinity ($S_p$) defined water masses Sea Ice Melt and Melt Water (SIM & MW), Upper Halocline Layer (UHL) and the Atlantic Layer (ATL) from PC1 of the original 8-variable PCA (Fig. 7) with exception to $^{226}Ra$, which does not significantly coincide with PC1.

| Water Mass | $S_p$ | DIC | AT | $\delta^{18}O$ | Ba | $^{226}Ra$ | $^{228}Ra$ |
|---|---|---|---|---|---|---|---|
| SIM & MW | 0 | 607 | 648 | -18.9 | 105 | 25.2 | |
| Shelf | 25 | | | | | | 22.4 |
| UHL | 33.1 | 2145 | 2282 | -1.18 | 50.8 | 10.6 | 5.3 |
| ATL | 35 | 2233 | 2375 | -0.157 | 47.7 | 9.8 | 1.3 |



Mears et al., Figure 1

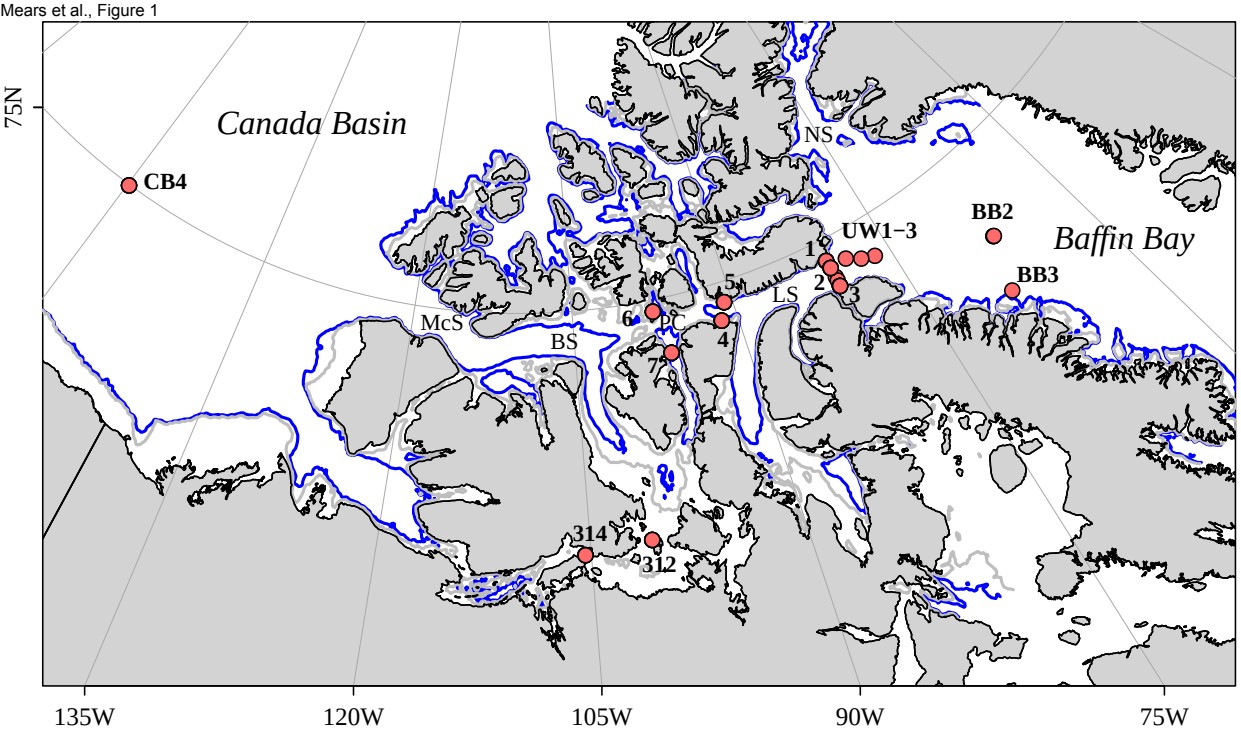





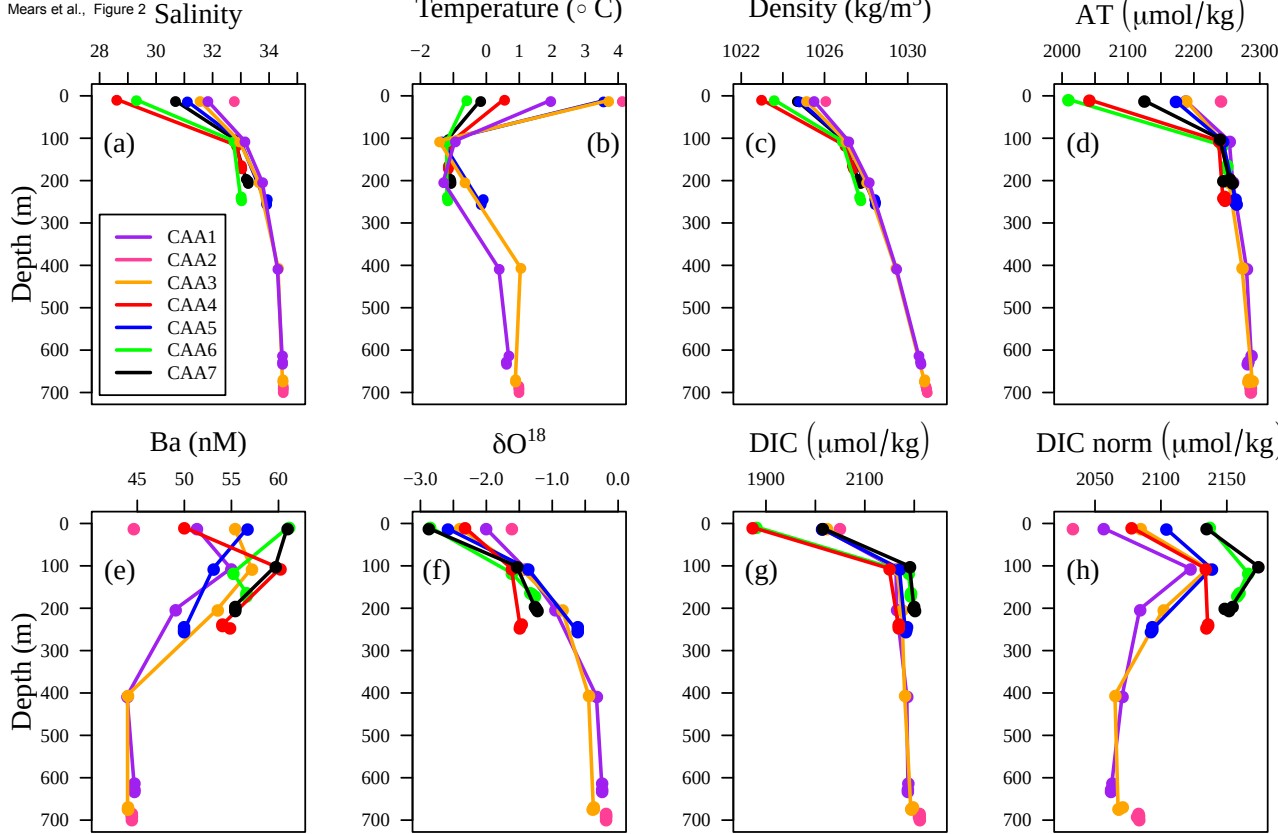





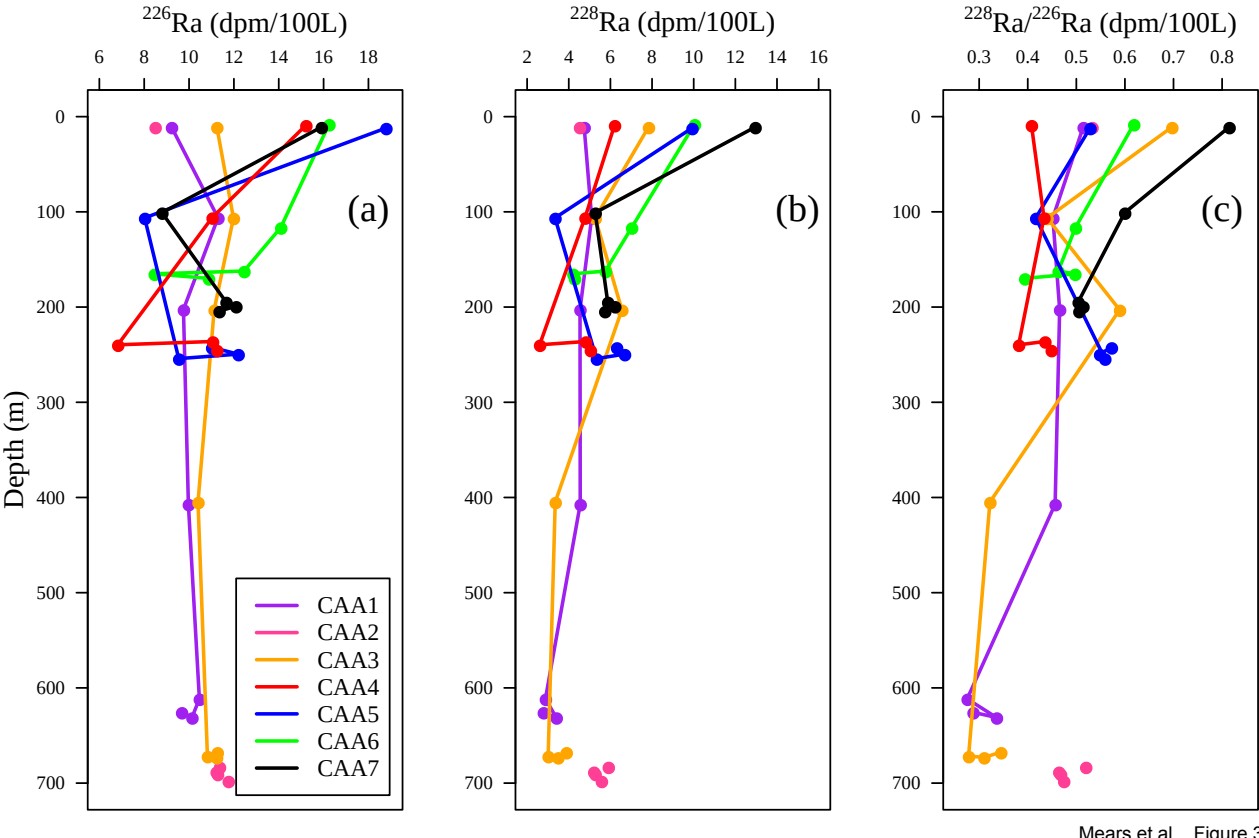





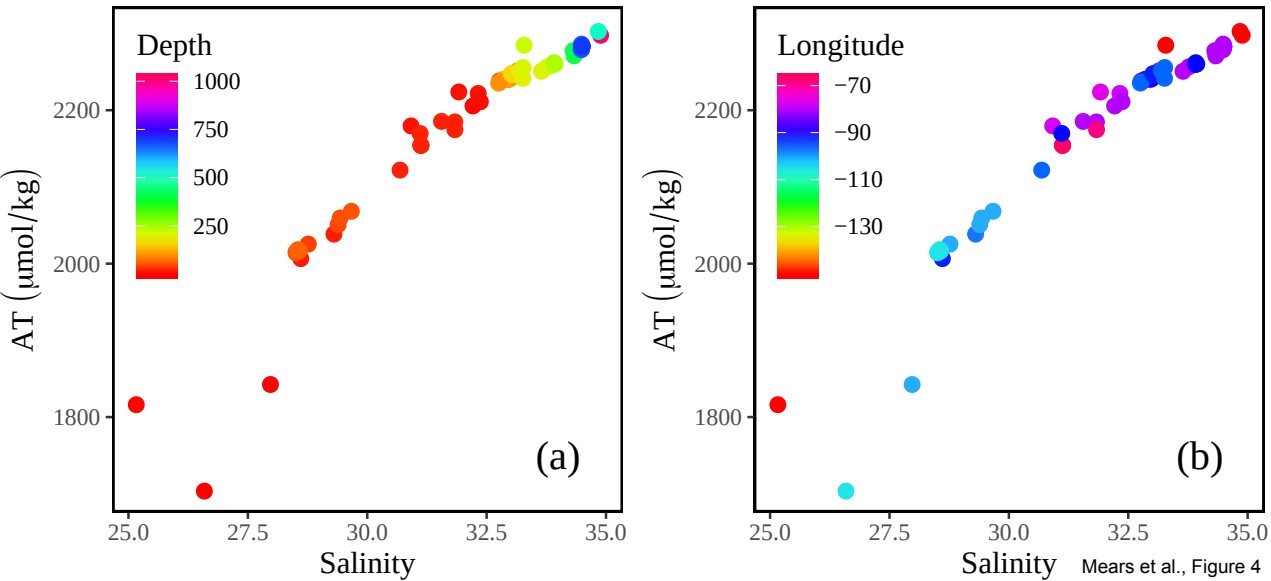

Mears et al., Figure 4





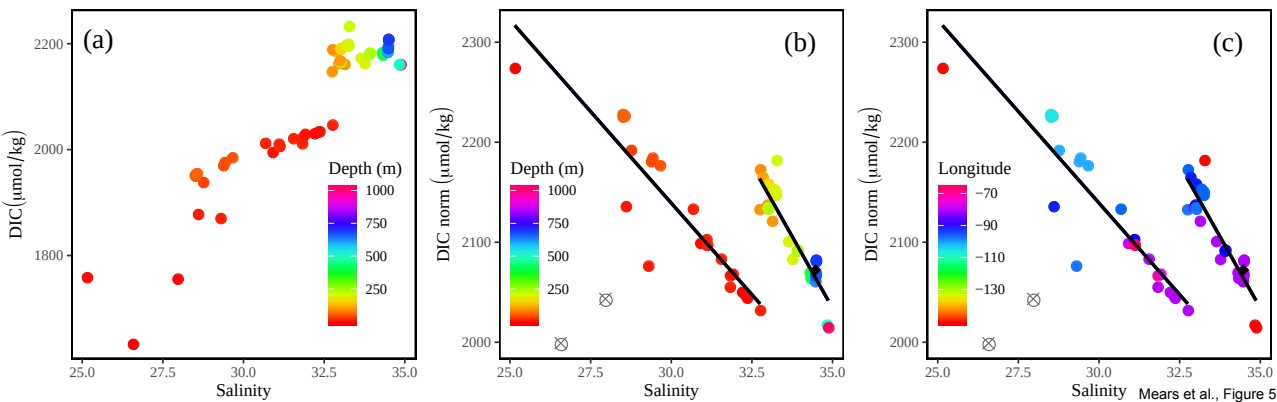

Mears et al., Figure 5



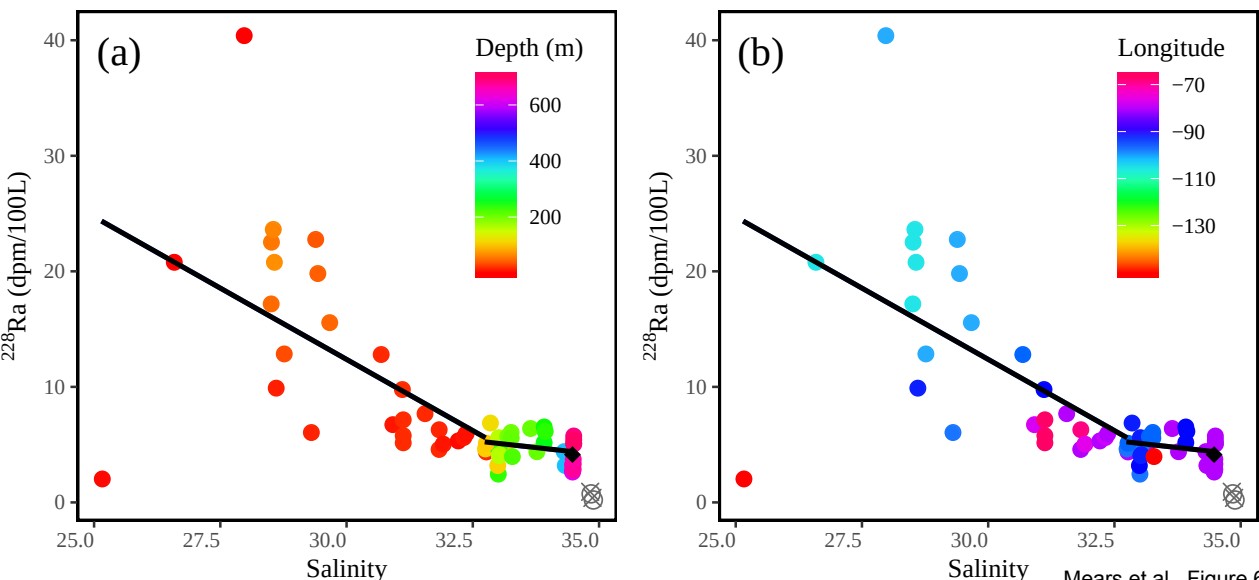

Mears et al., Figure 6





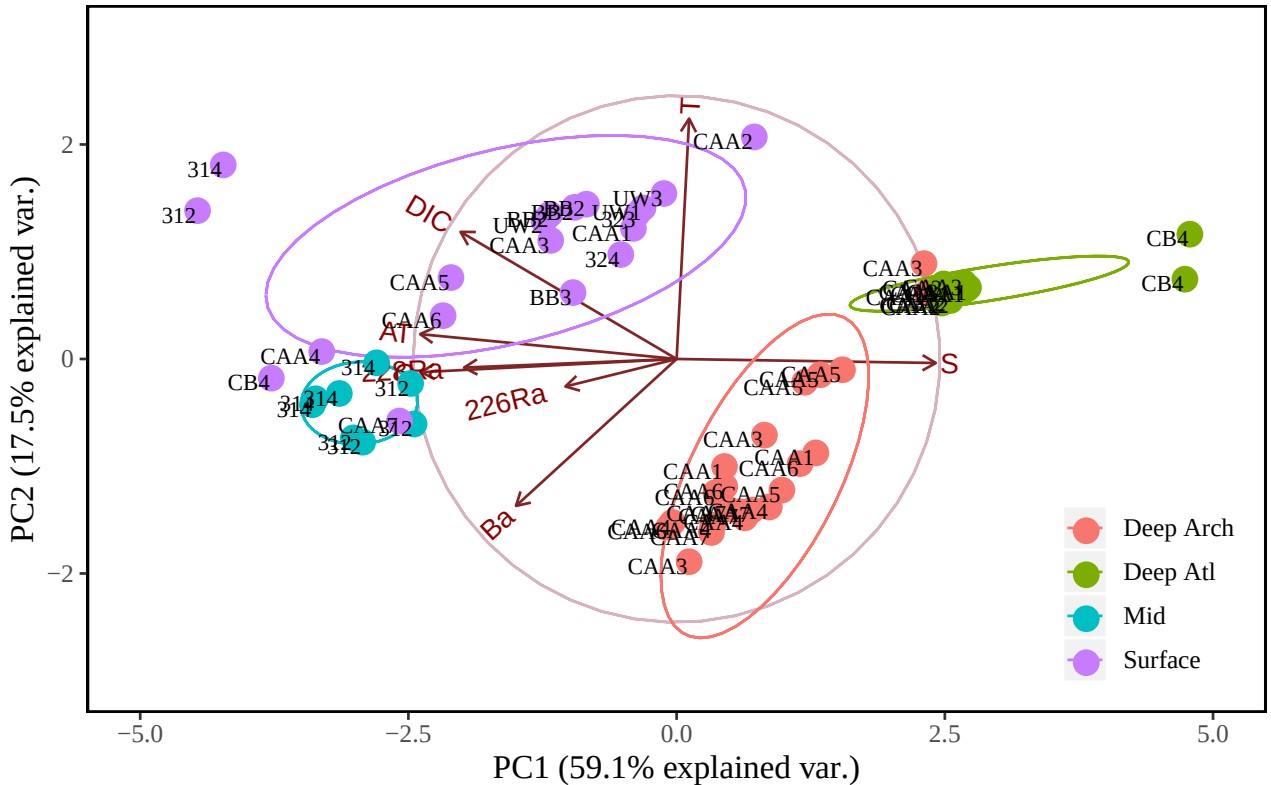

Mears et al., Figure 7



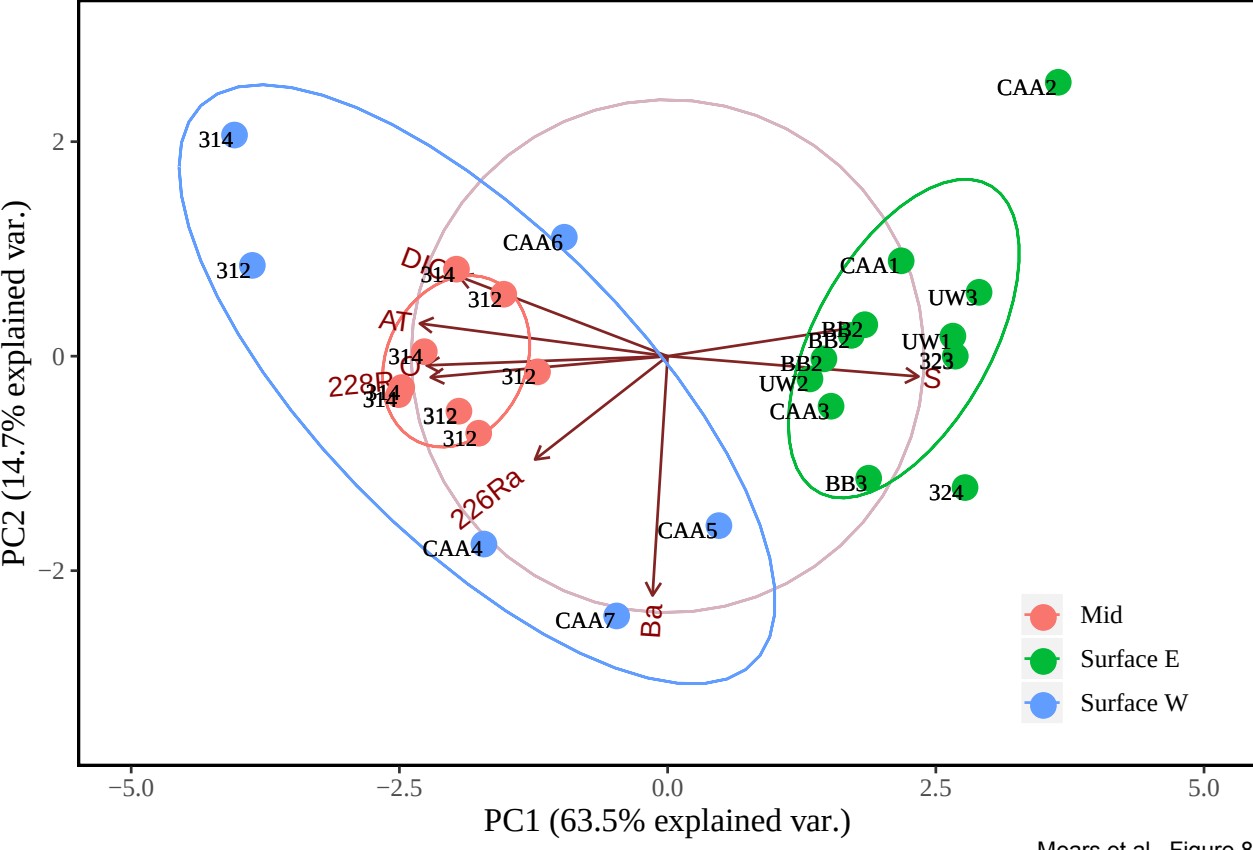

Mears et al., Figure 8





Mears et al., Figure 9





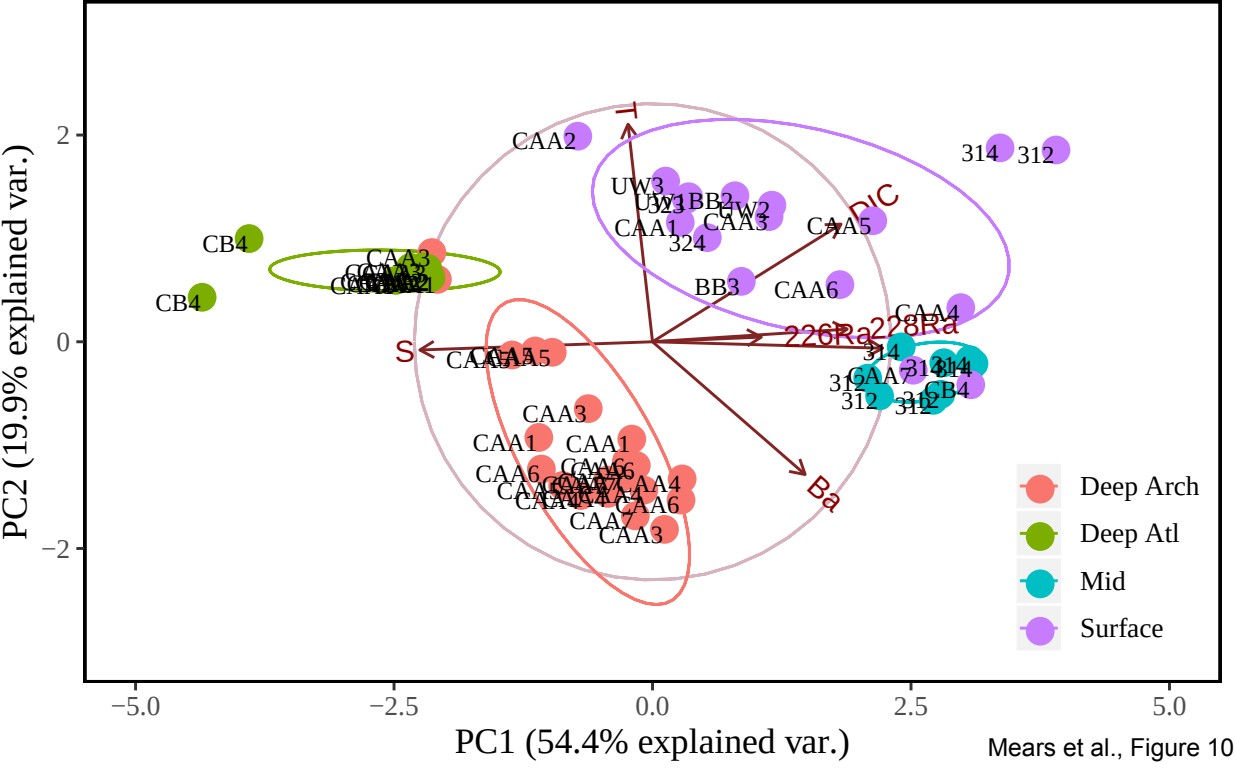





Mears et al., Figure 11



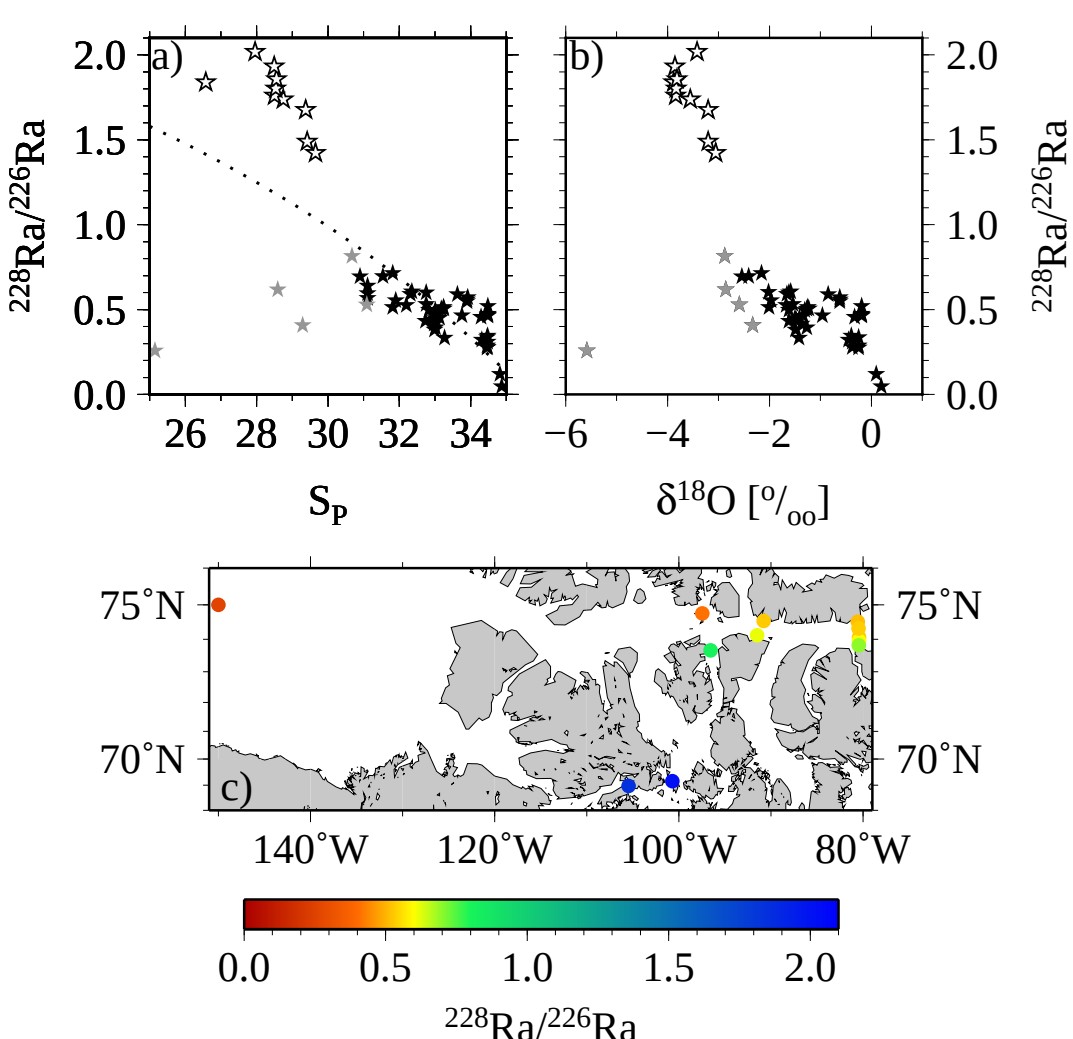

Mears et al., Figure 12





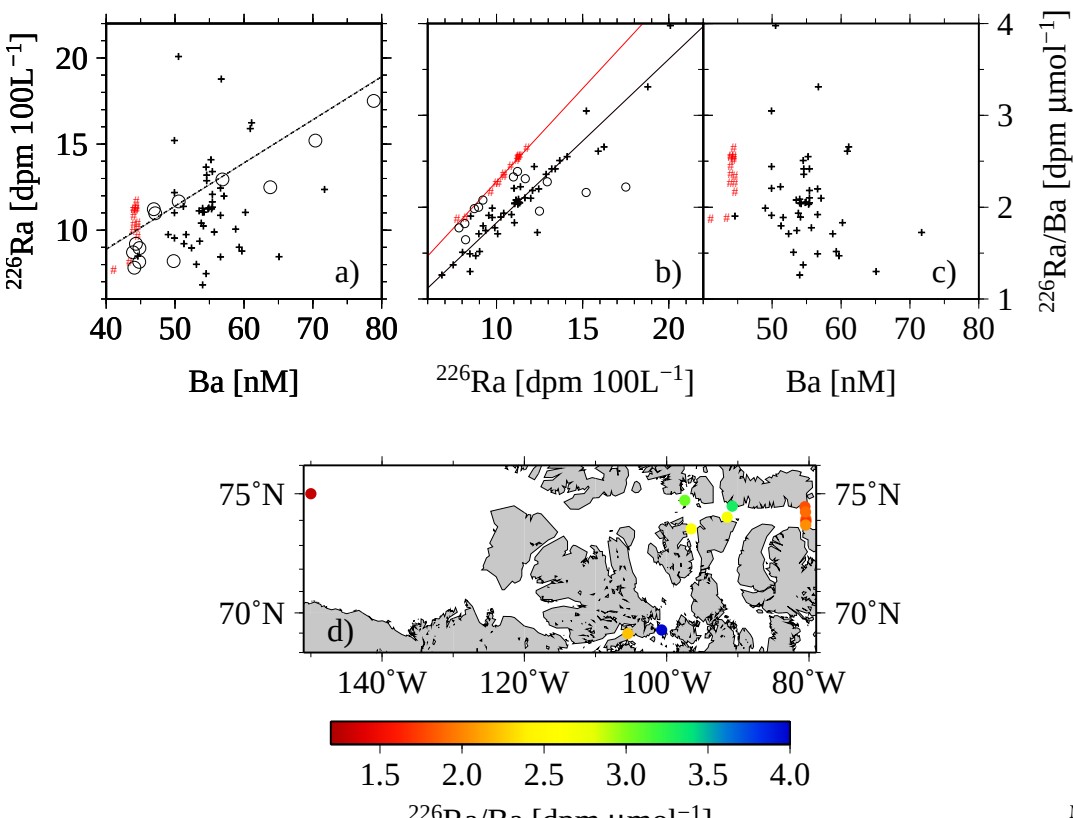

Mears et al., Figure 13





Mears et al., Figure 14

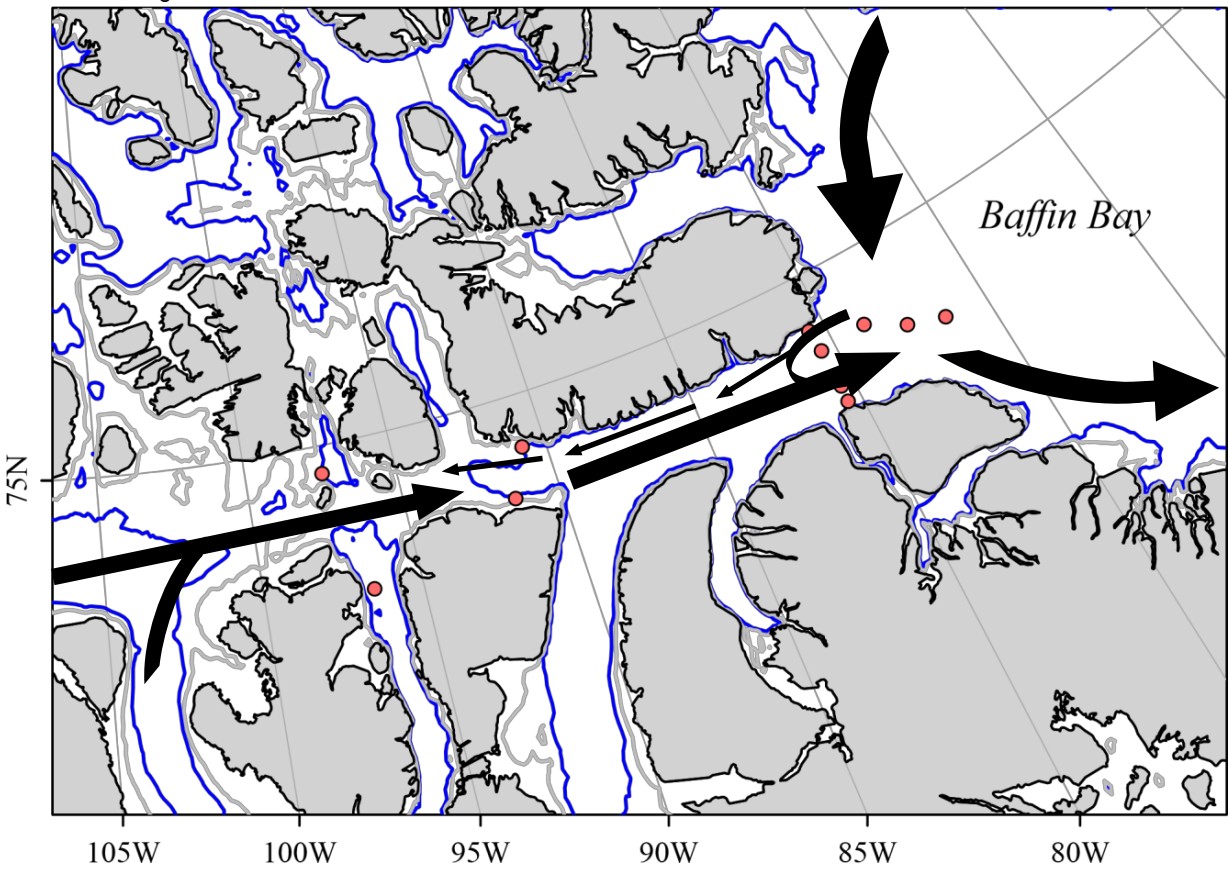