# Peer review of "Using 226Ra and 228Ra isotopes to distinguish water mass distribution in the Canadian Arctic Archipelago"

_Biogeosciences, 2020_

## Referee Comment (RC1) · Michiel Rutgers van der Loeff (Referee) · 15 Apr 2020

The paper presents and discusses the results of a study of long-lived radium isotopes in the Canadian Archipelago as part of a larger GEOTRACES study conducted 2015-2016 in the Arctic Ocean. The study is supported by parallel analyses of Ba and of the carbonate system (DIC, AT). The chemical and hydrographical data are subjected to a principal component analysis and the results are used to derive apparent water mass endmembers. The general features are well explained by 228Ra releases from (especially southern) shelf regions, a separation of water masses by the pycnocline (at about 100m depth), an eastward outflow of water from the Canada Basin over the sill in

Barrow Strait (less than 200m deep), and an anticlockwise recirculation of water from the Baffin Bay through the Lancaster Sound.

Graphs are very clear but a few are in my view not necessary for the message. Unless a clearer reason is given for the need for the normalization of DIC, Figs. 5 b and c can be omitted. Figs. 13 b and c do not add to the information given in Fig. 13a.

This paper adds an important chapter to the distribution and behavior of Radium isotopes and their use as water mass tracers in the Arctic Ocean. It deserves publication in Biogeosciences after a minor revision. It is generally well written, but there are several paragraphs that should be clarified.

Specific comments: 101: "westward" or eastward? 122: "nearly conservative". You might mention that there is a certain uptake (with Ba) by primary production 135: why is salinity called Sp? Why not just S as usual (and as used in eq 2)? 218: I understand that 228Ra was also regressed against salinity to derive the endmember value at S=25. 233: "similar results". It is not clear whether these authors found similar east-west gradients or similar exceptions to that rule as the example of CAA3 and CAA5. 251: I do not find CB4 results in Fig 2

246-256: this paragraph is too condensed to be understandable. I would like to see a bit clearer formulations. 246-247: For the intercept values of AT and DIC versus salinity, refer to Figs. 4 and 5a. I understand that the intercept in DIC is used later for the normalization of DIC. That makes it important to show the regression lines and indicate the error of the intercept values. 248: "thus attributed" I don't understand what is meant here 253-255: "similar results"? With respect to longitude yes, but without the metabolic maximum 254: "both"? 255: do you mean that at the depth of the pycnocline both S and AT but not DIC are increasing eastward?

Lines 263-274; Normalization of DIC: I understand from Friis et al. (2003) that normalization is meant to remove the salinity dependence of DIC, which is apparently not the result here (compare Figs. 5 a,b). The rationale of this normalization is not well explained. The difference between surface and subsurface water masses is clearly seen in Fig. 5a (especially when in panel a the symbols belonging to anomalous station CB4 are marked as they are in panel b and c) and is not made clearer by the normalization procedure depicted in Figs 5b and c. 272: "consistent" is misplaced here. The inverse longitudinal trend is noted in the next sentence, but it is not explained why the accumulated DIC decreases "as waters flow longitudinally eastward through the CAA". When surface water flows eastward, DIC increases, S increases and DICnorm decreases. Is this the non-conservative behavior meant in line 266 to be shown by the normalization (evasion of CO2?) or just an effect of mixing with high-S/low-DICnorm water from Baffin Bay? 363: why does the similarity of Stas CAA1, 3 and 5 to the Deep ATL group suggest a westward intrusion along the northern edge when Stas 1 and 5 are along the northern edge but Sta 3 is along the southern edge?

380: Explain why you show in Fig. 11d the 226Ra/Ba ratio and not 226Ra itself. Why is the scale in Fig. 11c inverted? That is a bit confusing.

Lines 386 and further; Figure 13: It is very useful to plot 226Ra vs Ba and to map the 226Ra/Ba ratio. But what is the advantage of plotting the ratio against each of its components 226Ra and Ba (Figs 13 b,c)? In 13a, what is the broken line? If this is the relationship found by Le Roy (and/or van Beek), then mention it. If not, then include it. What are the circles? They are missing in panel13c. The red symbols (S>34) in Fig. 13a appear to show that 226Ra is independent on Ba, in contrast to the findings of LeRoy and van Beek. In line 391 you may mean that the average position of the red symbols (∼10 dpm/100L; 44 nM) falls along that line, but their 226Ra variation is not associated with a corresponding variation in Ba.

396: Please give a reference for the high Ba runoff of rivers draining into the CAA.

Lines 476-509; Figure 12: What is the meaning of the broken line in 12a? Is that the relationship based on apparent endmembers? Perhaps this is related to the expressions "higher" and "lower . . . than the mixing ratios" in lines 487-494. Please

clarify. The symbols in Figs.12 a,b appear to have no other background than their position relative to each other on this graph. It is then more appropriate to use an enveloping ellipse as in Fig 9 rather than different symbols. It might be interesting to identify the outliers geographically (in Fig 12 c). Legend: flow paths? The figure is discussed again in lines 503-509, but that formulation must be improved. Line 502: you mean: the 228Ra/226Ra -$\delta$18O relationship (Fig. 12b). If you want to show non-conservative behavior from this relationship, shouldn't you draw an endmember mixing line as in panel a? Line 503-505: I guess you mean that conservative mixing of 228Ra or 228Ra/226Ra with a pure riverine source can be excluded, but that will hold for a plot against salinity as well as for a plot against 18O.

FIGURES: Fig. 7: make sure that the names of all the parameters are readable. "18O" is covered. Fig 14: Is this surface flow? If not specify at what depth.

Michiel Rutgers van der Loeff, 15 April, 2020

---

## Referee Comment (RC2) · Michael E. Böttcher (Referee) · 17 Apr 2020

The submitted manuscript on "Using 226Ra and 228Ra isotopes to distinguish water mass distribution in the Canadian Arctic Archipelago" by Mears, Thomas, Henderson, Charette, MacIntyre, Dehairs, Monnin, and Mucci, presents a detailed data set for a hydrographic and chemical characterization of water masses in the Canadian Arctic Archipelago that was derived during the GEOTRACE program: Oxygen isotopes (18O/16O) of water, radium isotopes, parameters of the dissolved carbonate system (AT, DIC), and dissolved Ba. The data are used to separate water masses and mixing properties, and define Ra sources. A thermodynamic analysis is used to defined

the saturation states of the aqueous solution wrt. barite, and all data are furthermore analyzed by a principle component analysis (PCA).

I will not repeat the arguments already provided in the detailed review by Michiel Rutgers van de Loeff, which I agree upon, and only concentrate on further minor aspects.

Overall, the data data set is original, impressive, and the conclusions derived from the investigation are well argued, including the presenting figures.

Detailed comments:

- I suggest to add a covariation diagram and further discussion of d18O-H2O values versus salinity.

- Fig.2: Its: d18O.

- Table 2: Units are missing.

- Reference list:

-L624: What kind of publication is this?

-e.g., lines 618, 631, 664, 710, 728 etc.: Please delete all informations about 'access dates' and if an article was read on-line from the reference list. Serious scientific journals guarantee that the scientific content published after acceptance is the same off- and on-line, and keeps its content in all details over time. This may be different for other on-line sources, that are often questionable permanent references.

Michael Ernst Böttcher, April 17th, 2020

---

## Referee Comment (RC3) · Amber Annett (Referee) · 23 Apr 2020

Mears and colleagues present an extensive dataset from the Arctic, spanning the Canada Basin to Baffin Bay via the Canadian Arctic Archeipelago, with the aim of characterizing water mass end-members, evolution of properties and transport. The study employs a range of parameters, including DIC, Ba, oxygen and Ra isotopes to probe the influence of freshwater and shelf interaction, detecting westward flow along the northern flank of Lancaster Sound.

Overall, the context and aims are well established. The introduction provides necessary background to the regional oceanography and methodological approaches. Methods are appropriate, adequately detailed and the results clearly support the conclusions drawn and are well situated within the existing literature. I agree with the comments already provided by reviewers Rutgers van der Loeff and Böttcher, including that the manuscript represents an addition worthy of publication in Biogeosciences after minor revisions. My additional comments are that the presentation of some results could be improved to make the interpretations more immediately visible from the figures (specified below), and that providing some additional quantitative details in the implications section could enhance the impact of this manuscript.

Line 101: The final sentence of this paragraph is not needed, and references to what will be discussed or unravelled later in the manuscript should be minimized where possible.

Line 373: Not substantially lower, unless you define the depths – Fig 11c indicates higher 228/226Ra ratios at 100m and 400m, with lower ratios only present at 0, 200 and 700m.

Line 380: the opposing gradients, do you mean the different strength of the gradient? Line 374 says "the same pattern"

Line 467: clarify the impact of longer circulation history (it could be interpreted as 228Ra decay, or accumulation of 228Ra from shelf inputs)

Line 296: Mention of CB4 could be integrated more clearly into this paragraph explicitly linking it to the southern flank of LS. Further, what level of confidence do you have in where the inflow turns back around or ceases to influence water properties? The westward flow is not shown reaching CAA6 in Fig 14, but the north/south gradient persists between CAA6 and CAA7 in Figs 12c and 13c - if there is uncertainty in how far this water reaches you could include a dashed arrow?

Line 525: "rerouted" rather than rerouting

Line 531: This section feels a bit unsupported; some quantification effort would deliver

meaningful context for using these findings as a tool to probe impacts or vulnerability to climate change, and increase the impact of the manuscript. Some suggestions: based on 228Ra decay, what is the minimum time scale of the eastward transport of water between CAA1 and CAA6? This must make some assumptions (e.g. no additional inputs) but would provide a minimum time scale; is it rapid or slow? What are the temperature differences between east and westward flowing waters? Where will increased heat be delivered – pumped into LS or out into Baffin Bay? Does any historical data support a strengthening or weakening of this u-turn route and what does/would that mean for transport of heat (or nutrients, or any other parameter).

Fig 2: If full CTD cast data is available, it would be preferable to show this rather than only the points for which DIC/Ra/Ba samples were collected, in order to situate samples relative to pycnoclines, thermal minima, etc.

Fig 4: Pink/red colours are difficult or impossible to distinguish for a few of the points. Stick to a different palette for clarity?

Fig 6 legend: Should this read Grey circles? If there are diamonds, they're not visible. Repeat here what the division is between "surface" and "deep" waters.

Fig 8: refer to section of text where the east/west boundary is stated (can it be shown on a map as well?)

Fig 9: Same comment regarding colour bar as Fig 4. Could the CB4 sample be designated with a different symbol so that it can be picked out easily on c and d?

Fig 11 & 12: Colour bar for 228/226Ra should also go from blue (low) to red (high) - otherwise the rationale for the inversion must be presented.

Fig 12: This figure panel needs overlap with the black/grey/white symbols of 12a and 12b, it's currently not possible to see where each group was collected from on the map. Also there's plenty of space on the map, please label 312 and 314 as readers may not remember all the station designations.

Fig 13: Include legend on the figure showing what red symbols denote.

[Figure]

---

## Editor Comment (EC1) · Manmohan Sarin (Editor) · 14 May 2020

This is a potentially interesting study on use of Radium isotopes to distinguish water mass distribution in the Canadian Arctic Archipelago.

Further to the comments made by three reviewers, authors' may like to provide some quantitative information (in the abstract) on the concentrations of radium isotopes and 228-Ra/226-Ra ratios measured in different water masses. There is no single result/number provided in the abstract.

Lines 48-51, in abstract, can be moved to conclusion/implication section.

[Figure]

Authors' may like to reassess the regression parameters (slope, intercept and R2) stated in figure captions for 5 and 6. Are these numbers significant and meaningful to 3rd and 4th decimal units? For example, in Figure 6, regression analysis is given as: y = -2.4666x + 86.377 (R2 = 0.2835) for the surface trend and y = -0.4854x +21.127 (R2 = 0.0722) for the deep trend. The slope, intercept and R2 values for linear regression analysis are rather absurd (3rd/4th decimal) considering the analytical uncertainties in the measurements of radium isotopes in individual water samples.

Line 109: For the benefit of a general reader, it may be relevant to name the parent isotopes of Radium, 230-Th (226-Ra) and 232-Th (228-Ra).

Line 161: Was there any attempt to simultaneously use 295 keV peak to quantify 226-Ra?

―――――――――――――――――――――――

---

## Author Comment (AC1) · 6 Jun 2020

**Response to Interactive Comments from: "Using 226Ra and 228Ra isotopes to distinguish water mass distribution in the Canadian Arctic Archipelago" by Chantal Mears et al. Michiel Rutgers van der Loeff (Referee).**

(Please note, the Referees comments are italicized and our responses are un-italicized.)

We would like to thank Michiel Rutgers van der Loeff for taking the time to comment in great detail on our paper.

*The paper presents and discusses the results of a study of long-lived radium isotopes in the Canadian Archipelago as part of a larger GEOTRACES study conducted 2015- 2016 in the Arctic Ocean. The study is supported by parallel analyses of Ba and of the carbonate system (DIC, AT). The chemical and hydrographical data are subjected to a principal component analysis and the results are used to derive apparent water mass endmembers. The general features are well explained by 228Ra releases from (especially southern) shelf regions, a separation of water masses by the pycnocline (at about 100m depth), an eastward outflow of water from the Canada Basin over the sill in Barrow Strait (less than 200m deep), and an anticlockwise recirculation of water from the Baffin Bay through the Lancaster Sound.*

*Graphs are very clear but a few are in my view not necessary for the message. Unless a clearer reason is given for the need for the normalization of DIC, Figs. 5 b and c can be omitted. Figs. 13 b and c do not add to the information given in Fig. 13a*

We thank the referee for this comment. Indeed, Figs 5b,c and 13b,c appear "underrepresented" in our discussion. Still, in our opinion these Fig 5b/c, i.e. $DIC_{norm}$, provides valuable understanding to the watermass distribution, as Fig 13b,c do by identifying species controlling the $^{226}Ra/Ba$ ratio. In any revised version we will aim to clarify and underpin the use to these panels.

*This paper adds an important chapter to the distribution and behavior of Radium isotopes and their use as water mass tracers in the Arctic Ocean. It deserves publication in Biogeosciences after a minor revision. It is generally well written, but there are several paragraphs that should be clarified.*

*Specific comments: 101: "westward" or eastward?*

Westward is intended here, but the sentence has been rephrased to "This suggests that there may be the intrusion of Atlantic waters originating from Baffin Bay, moving into the CAA along the northern edge from the east, and possibly creating a "U-turn" as the westward current reroutes back into Baffin Bay along the southern edge."

*122: "nearly conservative". You might mention that there is a certain uptake (with Ba) by primary production*

We agree with the referee and thus have changed this sentence to include Ra's role with biological uptake and accumulation at depth to "A slight enrichment can be seen in Pacific Ocean deep waters relative to deep water of the Atlantic Ocean, primarily due to $^{226}$Ra uptake by biology for the production of silicate or calcium tests or in barite ($BaSO_4$) co-precipitation (van Beek et al., 2007; Charette et al., 2015b; Moore and Dymond, 1991). With this exception in mind, $^{226}$Ra reveals a "nearly" conservative distribution in the oceans, thus facilitating its use as a long-term pelagic-based tracer of water masses and of shelf inputs (Broecker et al., 1967; Charette et al., 2015a; Chung, 1980),

*135: why is salinity called Sp? Why not just S as usual (and as used in eq 2)?*

In our understanding $S_P$ appears to be the new convention to refer to salinity.

*218: I understand that 228Ra was also regressed against salinity to derive the endmember value at S=25.*

We take note of the referees comment here, and thus have added "Lastly, linear regressions of each variable, with the exception of $^{228}$Ra, against the practical salinity were plotted to express robust end-member relationships from within the previously categorized salinity-defined water masses present throughout the CAA. For $^{228}$Ra, the end-member was derived from linear regression to the practical salinity where $S_P$=25, as it source is in shelf sediments, of which location is in the $S_P$ range of 25."

*233: "similar results". It is not clear whether these authors found similar east-west gradients or similar exceptions to that rule as the example of CAA3 and CAA5.*

To clarify the referees comment, the line has been adapted to "Prinsenberg and Bennett (1987) reported similar trend in results from samples collected in 1982 across Barrow Strait, a sill less than 200m deep located roughly between 105°W to 90°W, where analogous transects for salinity and temperature were recorded throughout the surface layer (Fig. 1)."

*251: I do not find CB4 results in Fig 2*

This station was purposefully left out of the profile comparison in Fig 2, as it's values differ greatly than those found within the Archipelago. Profiles of CB4 are provided in the Appendix, which has now references in line 251.

*246-256: this paragraph is too condensed to be understandable. I would like to see a bit clearer formulations.*

We agree with the Referee, the paragraph has been edited in hope to clarify the results: "The property-property diagrams of DIC and AT vs. S display strong positive relationships in surface waters (see Fig. 4 and 5a for regression intercepts and error). The surface waters of the stations west of 96.5°W thus appear impacted by sea-ice melt (SIM) and Meteoric Water (MW) (Figs. 1, 2, 4, 5c). Highest DIC concentrations were observed at the pycnocline of the western most station (CB4) (Fig. 2, Fig. 5a and Appendix 2). This maximum in metabolic (respiratory) DIC decreases slightly eastward due to the increasing contribution of low-DIC ATL waters (Fig. 5a) (Shadwick et al., 2011a). AT concentrations were found to increase more linearly with depth, void of the metabolic maximum witnessed within the DIC samples (Figs. 2d, g, 4). This is explained by the concomitant increase in AT and S values rather than metabolic activity, thus distinguishing AT from DIC (Burt et al., 2016a; Shadwick et al., 2011a; Thomas et al., 2011)."

*246-247: For the intercept values of AT and DIC versus salinity, refer to Figs. 4 and 5a. I understand that the intercept in DIC is used later for the normalization of DIC. That makes it important to show the regression lines and indicate the error of the intercept values.*

We agree with the referee and intercept values were removed from the text and referred to within the figures. Regression lines and error will be added to Figure 4 and 5a.

*248: "thus attributed" I don't understand what is meant here*

As discussed above, we have rewritten this section.

*253-255: "similar results"? With respect to longitude yes, but without the metabolic maximum*

As discussed above, we have rewritten this section.

*254: "both"?*

As discussed above, we have rewritten this section.

*255: do you mean that at the depth of the pycnocline both S and AT but not DIC are increasing eastward?*

As discussed above, we have rewritten this section.

*Lines 263-274; Normalization of DIC: I understand from Friis et al. (2003) that normalization is meant to remove the salinity dependence of DIC, which is apparently not the result here (compare Figs. 5 a,b). The rationale of this*

*normalization is not well explained. The difference between surface and subsurface water masses is clearly seen in Fig. 5a (especially when in panel a the symbols belonging to anomalous station CB4 are marked as they are in panel b and c) and is not made clearer by the normalization procedure depicted in Figs 5b and c.*

The normalization depicts here the clearly separated surface and deeper water masses. Both water bodies reveal individually conservative mixing gradients, but hardly, if any, mixing between them.

*272: "consistent" is misplaced here. The inverse longitudinal trend is noted in the next sentence, but it is not explained why the accumulated DIC decreases "as waters flow longitudinally eastward through the CAA". When surface water flows eastward, DIC increases, S increases and DICnorm decreases. Is this the non-conservative behavior meant in line 266 to be shown by the normalization (evasion of CO2?) or just an effect of mixing with high-S/low-DICnorm water from Baffin Bay?*

It is consistent in respect to the higher presence of meteoric waters in the western parts, which dilute the DIC of the PML on the one hand. The higher DICnorm reflects the high DIC concentration of the meteoric water, which is largely absent in the eastern parts. In the subsurface waters the observed decrease in DICnorm reflects the mixing of the Pacific waters (of western origin) with high respiratory DIC with the Atlantic waters (of eastern origin ) which carry a much lower respiratory DIC signal.

*363: why does the similarity of Stas CAA1, 3 and 5 to the Deep ATL group suggest a westward intrusion along the northern edge when Stas 1 and 5 are along the northern edge but Sta 3 is along the southern edge?*

We take into account the referees comment here and will aim to better explain the "U-turn" hypothesized, "Results of this analysis reveal that the Deep Arch stations CAA1, CAA3 and CAA5 are more closely linked to the Deep ATL group, implying that they are in fact part of the Deep ATL water mass (Fig. 10). This suggests that there is an intrusion of ATL water along the northern edge of the CAA. This westward flow with a speed of 2.2 cm/s was observed by Prinsenberg and colleagues (2009) and is weaker than the dominant eastward current flow (15.3 cm/s). This mild inflow of water along the northern edge of the Archipelago is then suspected to be redirected and exits back to Baffin Bay through the southern Archipelago station (CAA3)."

*380: Explain why you show in Fig. 11d the 226Ra/Ba ratio and not 226Ra itself. Why is the scale in Fig. 11c inverted? That is a bit confusing.*

Because of its long have life [226]Ra on its own does not provide a strong signal, in particular not in the deeper waters (Fig. 3). As [228]Ra is much more responsive to decay on the timescales relevant for this study the [228]Ra/[226]Ra ratio appears more appropriate to reveal systemic processes. We appreciate the note on the colour scale, this appears to be an error on our side.

*Lines 386 and further; Figure 13: It is very useful to plot 226Ra vs Ba and to map the 226Ra/Ba ratio. But what is the advantage of plotting the ratio against each of its components 226Ra and Ba (Figs 13 b,c)? In 13a, what is the broken line? If this is the relationship found by Le Roy (and/or van Beek), then mention it. If not, then include it. What are the circles? They are missing in panel13c. The red symbols (S>34) in Fig. 13a appear to show that 226Ra is independent on Ba, in contrast to the findings of LeRoy and van Beek.*

We apologize for this oversight: In Fig 13a,b the open circles are the data from van Beek et al., the dashed line in Fig 13a is redrawn from LeRoy et al. The regressions in Fig 13b are from our own data as specified in the caption. We will add this to the caption.

*In line 391 you may mean that the average position of the red symbols (~10 dpm/100L; 44 nM) falls along that line, but their 226Ra variation is not associated with a corresponding variation in Ba.*

Unfortunately, it is not fully clear to us, what the referee intends to state here? It was our intend to show that $^{226}$Ra and Ba reveal a somewhat "noisy" relationship within the CAA (Fig 13a). However, if you normalized the $^{226}$Ra to Ba, i.e., use the $^{226}$Ra/Ba ratio, it becomes evident that this ratio ins controlled by the variability in $^{226}$Ra and a "Ba-offset" originating from the CAA. This offset is not visible in the Atlantic data, neither in ours, nor in the ones plotted from literature.

*396: Please give a reference for the high Ba runoff of rivers draining into the CAA*

Yes we agree with the referee and this will be added to the paper (Gauy&Falkner 1998).

*Lines 476-509; Figure 12: What is the meaning of the broken line in 12a? Is that the relationship based on apparent endmembers? Perhaps this is related to the expressions "higher" and "lower . . . than the mixing ratios" in lines 487-494. Please clarify. The symbols in Figs.12 a,b appear to have no other background than their position relative to each other on this graph. It is then more appropriate to use an enveloping ellipse as in Fig 9 rather than different symbols. It might be interesting to identify the outliers geographically (in Fig 12 c). Legend: flow paths? The figure is discussed again in lines 503-509, but that formulation must be improved.*

We appreciate this comment. In essence, all information is given, but not well enough presented. The dashed line has been compiled from the data given in table 2, and the geographical attribution is shown by the shading of the symbols, but we have fallen short in spelling that out. We will improve the wording of that section.

*Line 502: you mean: the 228Ra/226Ra -δ18O relationship (Fig. 12b). If you want to show nonconservative behavior from this relationship, shouldn't you draw an endmember mixing line as in panel a?*

The issue with compiling the $\delta^{18}O$ mixing line is that the two freshwater endmembers (sea-ice and meteoric water) reveal different endmember characteristics. As discussed our analysis (Tab. 2) gives "only" an apparent, ie. mean endmember, which however is in reality a composite (see for example Thomas et al., 2011 and elsewhere). Instead, we intend to use that plot to unravel the characteristics of the two low-$S_P$ groups of $^{228}Ra/^{226}Ra$ samples, one with less negative $\delta^{18}O$, pointing to sea-ice melt and long history and distance from the $^{228}Ra$ source, and one imprinted by more negative $\delta^{18}O$ (meteoric water) from within the CAA with young history and proximity to the $^{228}Ra$ source. As indicated in the preceding section we will attempt to improve the respective wording.

*Line 503-505: I guess you mean that conservative mixing of 228Ra or 228Ra/226Ra with a pure riverine source can be excluded, but that will hold for a plot against salinity as well as for a plot against 18O.*

As indicated in the preceding section we will attempt to improve the respective wording.

*FIGURES: Fig. 7: make sure that the names of all the parameters are readable. "18O" is covered.*

We acknowledge the referees comment here and have made the parameter titles within Fig 7 visible.

*Fig 14: Is this surface flow? If not specify at what depth.*

We agree with the referees comment and the figure caption for Figure 14 was changed accordingly to "**Figure 14:** Sketch of proposed surface flow pattern as identified in the current study."
We would like to thank Michael E. Böttcher for taking the time to review our manuscript. Any comments provided by the Referee will be placed in italics while answers will be un-italicized.

*The submitted manuscript on "Using 226Ra and 228Ra isotopes to distinguish water mass distribution in the Canadian Arctic Archipelago" by Mears, Thomas, Henderson, Charette, MacIntyre, Dehairs, Monnin, and Mucci, presents a detailed data set for a hydrographic and chemical characterization of water masses in the Canadian Arctic Archipelago that was derived during the GEOTRACE program: Oxygen isotopes (18O/16O) of water, radium isotopes, parameters of the dissolved carbonate system (AT, DIC), and dissolved Ba. The data are used to separate water masses and mixing properties, and define Ra sources. A thermodynamic analysis is used to defined the saturation states of the aqueous solution wrt. barite, and all data are furthermore analyzed by a principle component analysis (PCA).*

*I will not repeat the arguments already provided in the detailed review by Michiel Rutgers van de Loeff, which I agree upon, and only concentrate on further minor aspects.*

*Overall, the data data set is original, impressive, and the conclusions derived from the investigation are well argued, including the presenting figures.*

*Detailed comments:*

*- I suggest to add a covariation diagram and further discussion of d18O-H2O values versus salinity.*

This has been shown and discussed in Thomas et al., 2011 their Fig. 5d. Please see also our comment in response to referee 1's comment re. line 502. We will add a respective panel as insert to Fig. 12b.

*- Fig.2: Its: d18O.*

We agree with the Referee, this was an over look on our part and has been changed within the figure.

*- Table 2: Units are missing.*

We agree with the Referee, this was an oversight on our part and has been changed within the table.

*Reference list: L624: What kind of publication is this?*

*-e.g., lines 618, 631, 664, 710, 728 etc.: Please delete all informations about 'access dates' and if an article was read on-line from the reference list. Serious scientific journals guarantee that the scientific content published after acceptance is the same offand on-line, and keeps its content in all details over time. This may be different for other on-line sources, that are often questionable permanent references.*

Thank you Referee, the proper citation has been placed for L624 and the access dates have been deleted from the reference list.

*Michael Ernst Böttcher, April 17th, 2020*

Thank to Amber Annett for taking the time to review our manuscript. The Referees comments are italicized while our answers are unitalicized.

*Mears and colleagues present an extensive dataset from the Arctic, spanning the Canada Basin to Baffin Bay via the Canadian Arctic Archeipelago, with the aim of characterizing water mass end-members, evolution of properties and transport. The study employs a range of parameters, including DIC, Ba, oxygen and Ra isotopes to probe the influence of freshwater and shelf interaction, detecting westward flow along the northern flank of Lancaster Sound.*

*Overall, the context and aims are well established. The introduction provides necessary background to the regional oceanography and methodological approaches. Methods are appropriate, adequately detailed and the results clearly support the conclusions drawn and are well situated within the existing literature. I agree with the comments already provided by reviewers Rutgers van der Loeff and Böttcher, including that the manuscript represents an addition worthy of publication in Biogeosciences after minor revisions. My additional comments are that the presentation of some results could be improved to make the interpretations more immediately visible from the figures (specified below), and that providing some additional quantitative details in the implications section could enhance the impact of this manuscript.*

*Line 101: The final sentence of this paragraph is not needed, and references to what will be discussed or unravelled later in the manuscript should be minimized where possible.*

Upon review we agree with the Referee and the last sentence of the paragraph "The importance of this "U-turn" will be discussed later in the results section", has been removed.

*Line 373: Not substantially lower, unless you define the depths – Fig 11c indicates higher 228/226Ra ratios at 100m and 400m, with lower ratios only present at 0, 200 and 700m.*

We agree with the Referee here, providing a depth is both necessary and adds clarity, the statement has then been changed to " Furthermore, substantially lower values of the $^{228}$Ra/$^{226}$Ra ratio at depth (Fig. 11c) indicate the inflow of Atlantic water on the northern side of Lancaster Sound as well as its outflow along its southern side".

*Line 380: the opposing gradients, do you mean the different strength of the gradient?*

We agree with the referees comment, this makes the discuss difficult to follow and has thus be changed to "Since $^{226}$Ra activities reveal a much larger north-to-south gradient across Lancaster Sound than Ba does (Figs. 2e, 3a), the discrepancy in strength of the gradients shown in Figs. 11c and 11d are dominated by changes in $^{226}$Ra."

*Line 374 says "the same pattern"*

We agree with the referee here and have changed the wording to better complement the statement to "The corresponding pattern is revealed with $^{226}$Ra/Ba"

*Line 467: clarify the impact of longer circulation history (it could be interpreted as 228Ra decay, or accumulation of 228Ra from shelf inputs)*

In essence it should be both. Data from van der Loeff (2003) are from the Eurasian Shelf thus close to a $^{228}$Ra source. The Atlantic waters in the CAA have been without shelf contact for longer times allowing for tangible decay of $^{228}$Ra. We will amend wording here.

*Line 296: Mention of CB4 could be integrated more clearly into this paragraph explicitly linking it to the southern flank of LS. Further, what level of confidence do you have in where the inflow turns back around or ceases to influence water properties? The westward flow is not shown reaching CAA6 in Fig 14, but the*

*north/south gradient persists between CAA6 and CAA7 in Figs 12c and 13c - if there is uncertainty in how far this water reaches you could include a dashed arrow?*

Thank you for this hint. This pattern should be attributed to the intrusion of the Northwest Passage waters, which we have drawn solely at one point. We will amend the figure accordingly.

*Line 525: "rerouted" rather than rerouting*

We agree with the Referee, and this has been changed within the manuscript.

*Line 531: This section feels a bit unsupported; some quantification effort would deliver meaningful context for using these findings as a tool to probe impacts or vulnerability to climate change, and increase the impact of the manuscript. Some suggestions: based on 228Ra decay, what is the minimum time scale of the eastward transport of water between CAA1 and CAA6? This must make some assumptions (e.g. no additional inputs) but would provide a minimum time scale; is it rapid or slow? What are the temperature differences between east and westward flowing waters? Where will increased heat be delivered – pumped into LS or out into Baffin Bay? Does any historical data support a strengthening or weakening of this u-turn route and what does/would that mean for transport of heat (or nutrients, or any other parameter).*

Thank you, we will amend this sentence to have a stronger relation to our paper.

*Fig 2: If full CTD cast data is available, it would be preferable to show this rather than only the points for which DIC/Ra/Ba samples were collected, in order to situate samples relative to pycnoclines, thermal minima, etc.*

We will inquire at the GEOTRACES data center about this proposition.

*Fig 4: Pink/red colours are difficult or impossible to distinguish for a few of the points. Stick to a different palette for clarity?*

We agree with the referee's an attempt will be made to change the colour bar to be discernable.

*Fig 6 legend: Should this read Grey circles? If there are diamonds, they're not visible. Repeat here what the division is between "surface" and "deep" waters.*

This is correct, it should have read grey circles and the characterization of surface or deep water masses has been repeated.

*Fig 8: refer to section of text where the east/west boundary is stated (can it be shown on a map as well?)*

We agree with the Referee, that it is added information and clarity to define what surface east and west are, and thus the definition has been added to the figure title of Figure 8. An additional map has not been made as there is a 10° longitudinal gap between stations west and east and therefore with reference to Figure 1, it is easily discernable.

*Fig 9: Same comment regarding colour bar as Fig 4. Could the CB4 sample be designated with a different symbol so that it can be picked out easily on c and d?*

We agree with the referee, an attempt will be made to highlight CB4.

*Fig 11 & 12: Colour bar for 228/226Ra should also go from blue (low) to red (high) - otherwise the rationale for the inversion must be presented.*

As responded to referee 1, this appears to be an error on our side. Thank you.

*Fig 12: This figure panel needs overlap with the black/grey/white symbols of 12a and 12b, it's currently not possible to see where each group was collected from on the map. Also there's plenty of space on the map, please label 312 and 314 as readers may not remember all the station designations.*

*Thank you, we will colour-code symbols in Fig 12a,b. Addressing hereby the related concern of referee 1 as well.*

*Fig 13: Include legend on the figure showing what red symbols denote*
This has already been indicated in the caption, but we will add information to the panel.
We would like to extend thanks to the editor, Manmohan Sarin for taking the time to provide comments. Below italicized are the editors comments, while un-italicized are our response.

*This is a potentially interesting study on use of Radium isotopes to distinguish water mass distribution in the Canadian Arctic Archipelago. Further to the comments made by three reviewers, authors' may like to provide some quantitative information (in the abstract) on the concentrations of radium isotopes and 228-Ra/226-Ra ratios measured in different water masses. There is no single result/ number provided in the abstract.*

Thank you, we will provide such information in the revised abstract, where appropriate.

*Lines 48-51, in abstract, can be moved to conclusion/implication section.*

This line is restated in the conclusion section. But draws importance to the relevance of the study and other studies comparable to ours.

*Authors' may like to reassess the regression parameters (slope, intercept and R2) stated in figure captions for 5 and 6. Are these numbers significant and meaningful to 3rd and 4th decimal units? For example, in Figure 6, regression analysis is given as: y = -2.4666x + 86.377 (R2 = 0.2835) for the surface trend and y = -0.4854x +21.127 (R2 = 0.0722) for the deep trend. The slope, intercept and R2 values for linear regression analysis are rather absurd (3rd/4th decimal) considering the analytical uncertainties in the measurements of radium isotopes in individual water samples.*

We agree with the Editor, these numbers digits are exaggerated and will be shorted for conciseness.

*Line 109: For the benefit of a general reader, it may be relevant to name the parent isotopes of Radium, 230-Th (226-Ra) and 232-Th (228-Ra).*

We agree with the editor, it could be helpful for the reader to have the Thorium parent isotope and thus the sentence has added this information in, "Additionally, both long-lived Ra isotopes are formed from the decay of different Thorium (Th) isotopes ($^{226}$Ra is the daughter of $^{230Th}$, while $^{228}$Ra is the daughter of $^{232}$Th) in sediments and are distributed to the ocean through porewater advection and diffusion across the sediment-water interface, primarily along coastlines or the bottom boundary layer (Charette et al., 2015)."

*Line 161: Was there any attempt to simultaneously use 295 keV peak to quantify 226- Ra?*

We have not attempted to use the 295KeV peak to quantify $^{226}$Ra, as the uncertainty of the $^{226}$Ra data would have been improved only incrementally, deemed insignificant for the purpose of the study.

---

## Editor Decision (ED1)

**Authors should consider making following corrections in the manuscript:**

**Abstract, Page 2, line 31:** Sentence should read as:
"As the two long-lived isotopes of the Radium Quartet, ----------". Both isotopes (226-Ra & 228-Ra) are not longest-lived.

**Abstract, Page 2, line 49:**
Which "radioactive isotopes" authors' are referring to? Reference can be made only to Radium isotopes as used in this study. Or else, authors' can prefer to write "radioactive tracers".

**Table 2:** It appears that authors' have not honoured their own corrections made in Manuscript Version 2. The activities of 226-Ra and 228-Ra (dpm/100L) are again given to $2^{nd}$ and $3^{rd}$ decimal units (25.181, 9.76, 5.28 and 1.27) in the final version.

**Figure Captions 5 & 6:** Are the intercept values given as 2680.9 and 3282.2 (Fig. 5) realistic? The significance of the numbers is equally important in addition to regression analysis. Likewise, the value of slope given as – 0.485x (Fig. 6) cannot be justified considering analytical uncertainties associated with the measured 228-Ra activity (dpm/100L).

**Interactive comments from Referee #3** *(Amber Annett)***:**
Authors have not adequately revised the text in response to the following comments of the Referee #3:

**General comment: "***My additional comments are that the presentation of some results could be improved to make the interpretations more immediately visible from the figures (specified below)*, **and that providing some additional quantitative details in the implications section could enhance the impact of this manuscript.***"*

**Specific comment, Line 531:** *"This section feels a bit unsupported; some quantification effort would deliver meaningful context for using these findings as a tool to probe impacts or vulnerability to climate change, and increase the impact of the manuscript. Some suggestions: based on 228Ra decay, what is the minimum time scale of the eastward transport of water between CAA1 and CAA6? This must make some assumptions (e.g. no additional inputs) but would provide a minimum time scale; is it rapid or slow?* **What are the temperature differences between east and westward flowing waters? Where will increased heat be delivered – pumped into LS or out into Baffin Bay? Does any historical data support a strengthening or weakening of this U-turn route and what does/would that mean for transport of heat (or nutrients, or any other parameter)".***

Authors should address to some historical perspective in support of their results/findings as pointed out by the referee (strengthening/weakening of U-turn route and transport of heat/nutrients). Overall, conclusion/implications section needs some quantitative information with regard to time scale of water transport and temperature differences between east and west flowing waters (as per Referee's comment).

---

## Author Response (AR2)

Answers to the Editors Comments:

We would like to thank the editor for taking the time to go through our paper again. Below the comments from the referees and editor are italicized and the location of the comment bolded, while the authors response is written in plain text.

**Abstract, Page 2, line 31:**

*"As the two long lived isotopes of the Radium Quartet". Both isotopes (226-Ra and 228-Ra) are not the longest lived.*

Although $^{226}$Ra and $^{228}$Ra are both the two most long lived Ra isotopes within the Quartet and the two most long lived Ra isotopes, we do understand the editors confusion and to clarify for the reader, the sentence was changed to "As the two long-lived isotopes of the Radium Quartet, $^{226}$Ra and $^{228}$Ra ($^{226}$Ra, $t_{1/2}$=1600y and $^{228}$Ra, $t_{1/2}$=5.8y) can provide insight into the water mass compositions, distribution patterns, as well as mixing processes and the associated timescales throughout the Canadian Arctic Archipelago (CAA)."

**Abstract, Page 2 line 49.**

*Which "radioactive isotopes" authors' are referring to? Reference can be made on to Radium isotopes as used in this study. Or else, authors' can prefer to write "radioactive tracers".*

Yes, we were addressing to the Radium isotopes in addition to the oxygen isotope used throughout the study. The aim of this more elusive language was to connection to the GEOTRACES tracer initiative; thus, the latter was chosen, replacing radioactive isotopes with "radioactive tracers".

**Table 2:**

*It appears that the authors have not honoured their own corrections made in Manuscript Version 2. The activities of 226-Ra and 228-Ra are again given to $2^{nd}$ and $3^{rd}$ decimal units.*

We appreciate this being noticed; one decimal place has now been used throughout.

**Figure Captions 5 & 6:**

*Are the intercept values given as 2680.9 and 3282.2 (Fig 5) realistic? The significance of the number is equally important in addition to regression analysis. Likewise, the value of the slope given -0.485x (Fig. 6) cannot be justified considering analytical uncertainties associated with the measured 228-Ra activity (dpm/100L).*

To account for each values significance, the standard error was calculated for each regression. For Figure 5 in the DIC by S relationship yielded, DIC=(53.4±2.75)*$S_p$+371.4±88.8 ($R^2$ = 0.862). Where similarly for the surface samples in Figure 5 Normalized DIC by Salinity, $DIC_{norm}$=- (19.7±3.92)*$S_p$+2680 ±120 ($R^2$=0.533) and at depth; $DIC_{norm}$=(-34.2±3.47)*$S_p$+3282 ±117 ($R^2$=0.765). The same was then done for Figure 6, where the shallow samples with f(x) = -2.47x + 86.4 ($R^2$ = 0.284) was rewritten to accommodate the error as f(x) = (-2.47± 0.784)x + 86.4±23.6 ($R^2$ = 0.284). And again, the deeper samples yielding, f(x) = (-0.485±0.312)x + 21.1±10.5 ($R^2$ = 0.072).

Furthermore, in text this was rewritten as:

"**Figure 5:** Dissolved Inorganic Carbon (DIC) (a) and salinity-normalized Dissolved Inorganic Carbon ($DIC_{norm}$) (b) were plotted against salinity ($S_p$), with colours indicating depth (m) (a, b) and longitude (c).

The DIC vs. $S_p$ regression yields DIC=$(53.4\pm2.76)*S_p+371\pm88.8$ ($R^2 = 0.8621$). The DIC$_{norm}$ vs. $S_p$ plots were fitted with a piecewise regression analysis representing the surface, DIC$_{norm}$=$(-19.7\pm3.92)*S_p+2680.9\pm120$ ($R^2=0.533$) and at depth; DIC$_{norm}$=$(-34.2\pm3.47)*S_p+3282\pm117$ ($R^2=0.765$). In plots b & c CB4 was excluded from the piecewise regression (represented by unfilled, crossed out grey circles), whereas stations 312 and 314 surface samples were excluded entirely. The black diamonds identify the average Atlantic deep-water samples from stations CAA1, CAA2 and CAA3.

**Figure 6:** $^{228}$Ra (dpm/100L) plotted against practical salinity with colour indicating depth (a) and longitude (b) fitted with a piecewise regression excluding the deep stations of the Canada Basin (grey circles) and yielding $^{228}$Ra = $(-2.47\pm0.784)*S_p + 86.4\pm23.6$ ($R^2 = 0.284$) for the surface trend (0-80m) and $^{228}$Ra = $(-0.485\pm0.312)*S_p + 21.1\pm10.5$ ($R^2 = 0.072$) for the deep trend (>80m). The average of Atlantic deep waters sampled from stations CAA1, CAA2 and CAA3 is defined by a black diamond."

**Interactive Comments from Referee 3 (Amber Annett): the text has not been adequately revised to respond to these comments.**

**General Comment:** *"My additional comments are that the presentation of some results could be improved to make the interpretations more immediately visible from the figures (specified below) and that providing some additional quantitative details in the implications section could enhance the impact of this manuscript.*

**Specific Comment, Line 531: "***This section feels a bit unsupported; some quantification effort would deliver meaningful context for using these findings as a tool to probe impacts or vulnerability to climate change, and increase the impact of the manuscript.*

We agree with the reviewer, it was brought to our attention that the link to climate change was unsupported throughout the paper as climate change was only mentioned in the introduction and conclusion. Thus, referencing to climate change had been removed from both these sections. Changing the last sentence in the originally submitted document to "Furthermore, this study provides an additional tool to better understand and characterize water mass distributions, flow patterns, mixing and their respective time scales in challenging sampling areas such as the Arctic." in our last submission.

*Some suggestions: based on 228Ra decay, what is the minimum time scale of the eastward transport of water between CAA1 and CAA6? This must make some assumptions (rg. No additional inputs) but would provide a minimum time scale; is it rapid or slow?*

We appreciate the reviewer's suggestion, this was an interest of the authors early on in the data analysis as well, although the timescale of water flow throughout the CAA is much faster than that of the decay of $^{228}$Ra. Thus, looking at the transport time between stations using the $^{228}$Ra does not allow us to shed light on the timescale of water flow accurately. As we describe in section III.III.I.II, transport speeds are on the order to 2cm-15cm/s, which would equal to a transport time of a few weeks between CAA and

CAA7, which cannot be resolved with [228]Ra given the half live of 5.8y. Rather, as we show the [228]Ra reveals the different sources of he watrer, which are much further away and can be resolved.

*What are the temperature differences between east and westward flowing waters? Where will increased heat be delivered -pumped into LS or out into Baffin Bay?*

To better understand distinct differences in the westward intruding waters into the CAA and that of the bulk eastward transport an additional study should be done. Although overall trends in the Temperature can be seen within this data set (see Figure 2), a more in-depth extrapolation cannot be verified within this data set due to the lack of surface samples, in particular within the region of the westward flowing water mass. Furthermore, the authors feel that no further scientific understanding can be drawn distinguishing temperatures from this dataset and an additional investigation would need to be done to better understand these flow patterns.

*Does any historical data support a strengthened or weakening of this U-turn route and what does/would that meant for transport of heat (or nutrients, or any other parameter).*

We appreciate the reviewer question, to the authors knowledge, mention of the "U-turn" in the literature has primarily only included it's positioning and distribution not the implications of such a flow. Again, the authors feel that this is an area for more research, measuring the westward intruding water more carefully.

***Authors should address to some historical perspective in support of their results/findings as pointed out by the referee (strengthening/weakening of U-turn route and transport of heat/nutrienst). Overall, conclusion/implications section needs some quantitative information in regard to time scale of water transport and temperature difference between east and west flowing waters (As per Referees comment).***

Given the design of the overall GEOTRACES study within the CAA**,** these detailed points, though appreciably relevant, can hardly be answered based on our data set, as sampling density appears insufficient to resolve any change from local/spatial variability at the time of sampling. Due to the large water volume (and thus wire-time) required for Ra and also other tracers, sampling density needed to be lower in return. We feel that the strength of the Ra as tracer lies in the identification of patterns as we show in our paper, while in depth-quantification of such patterns at times needs to be achieved using other means. Obviously, we were able to make such statements, however we do feel that you data are would be over-interpreted in doing so.

[revised manuscript text omitted]